# An electronic origin of charge order in infinite-layer nickelates

Hanghui Chen [1,2] ✉, Yi-feng Yang [3,4,5] ✉, Guang-Ming Zhang [6,7] ✉ & Hongquan Liu[1]

A charge order (CO) with a wavevector $\mathbf{q} \simeq (\frac{1}{3}, 0, 0)$ is observed in infinite-layer nickelates. Here we use first-principles calculations to demonstrate a charge-transfer-driven CO mechanism in infinite-layer nickelates, which leads to a characteristic $Ni^{1+}$-$Ni^{2+}$-$Ni^{1+}$ stripe state. For every three Ni atoms, due to the presence of near-Fermi-level conduction bands, Hubbard interaction on Ni-$d$ orbitals transfers electrons on one Ni atom to conduction bands and leaves electrons on the other two Ni atoms to become more localized. We further derive a low-energy effective model to elucidate that the CO state arises from a delicate competition between Hubbard interaction on Ni-$d$ orbitals and charge transfer energy between Ni-$d$ orbitals and conduction bands. With physically reasonable parameters, $\mathbf{q} = (\frac{1}{3}, 0, 0)$ CO state is more stable than uniform paramagnetic state and usual checkerboard antiferromagnetic state. Our work highlights the multi-band nature of infinite-layer nickelates, which leads to some distinctive correlated properties that are not found in cuprates.

Motivated by cuprates, superconductivity was proposed[1,2] and has been recently discovered in hole-doped infinite-layer nickelates that contain $Ni^{1+}$ ions of the similar $d^9$ electronic configuration[3–9]. While this seems to confirm the initial expectation and support the Mott scenario for high-temperature superconductivity[10], the nickelates show very different properties even in their undoped parent compounds. Instead of an antiferromagnetic Mott insulator, the resistivity of the parent compounds exhibits metallic behavior at high temperatures and an upturn below around 70 K[3,8,11,12]. So far, no static long-range magnetic order has been observed down to the lowest measured temperature[13,14], although spin fluctuations have been reported in experiment[15]. These are all different from cuprates and have stimulated heated debate concerning the nature of the nickelate parent state, from which superconductivity is born upon hole doping. Although several theoretical scenarios[16–20] have been proposed to emphasize the key role of single-band Hubbard interaction, a growing consensus is that multi-band physics is indispensable in nickelates[21–32]. What has not been sufficiently explored is whether this

multi-band feature combined with a strong correlation on Ni-$d$ orbitals may lead to some unique experimental consequences in infinite-layer nickelates.

Recently, a charge order (CO) with broken translational symmetry has been reported independently by three experimental groups in the parent compounds of nickelate superconductors[33–35]. This raises a question concerning the CO origin and its competition with other phases. CO in cuprates is well known and has been intensively studied[36,37]. Despite the seeming similarity, the CO in nickelates has a few important differences[33–35]. First, it already exists and has the highest onset temperature in undoped compounds. Second, unlike cuprates where doped holes reside on oxygen atoms, holes are mainly introduced into Ni-$d$ orbitals, as supported by the experiments[38,39]. Finally and most importantly, it has an ordering wavevector $\mathbf{q} \simeq (\frac{1}{3}, 0, 0)$, compatible with a commensurate modulation with period $3a_0$ ($a_0$ is the lattice constant). This stripe pattern has not been observed in cuprates[40]. Hence, these differences may imply a new mechanism for the CO in the nickelates, which is most

[1]NYU-ECNU Institute of Physics, NYU Shanghai, Shanghai 200122, China. [2]Department of Physics, New York University, New York, NY 10012, USA. [3]Beijing National Laboratory for Condensed Matter Physics and Institute of Physics, Chinese Academy of Sciences, Beijing 100190, China. [4]University of Chinese Academy of Sciences, Beijing 100190, China. [5]Songshan Lake Materials Laboratory, Dongguan, Guangdong 523808, China. [6]State Key Laboratory of Low-Dimensional Quantum Physics and Department of Physics, Tsinghua University, Beijing 100084, China. [7]Frontier Science Center for Quantum Information, Beijing 100084, China. ✉e-mail: hanghui.chen@nyu.edu; yifeng@iphy.ac.cn; gmzhang@tsinghua.edu.cn

probably associated with the multi-band nature of the parent compounds[21–32].

In this work, we first use density functional theory plus dynamical mean field theory (DFT+DMFT) calculations[41–44] to reveal a special charge-transfer-driven CO mechanism in a prototypical infinite-layer nickelate $NdNiO_2$. We find that for every three Ni atoms, due to the presence of Nd-$d$/interstitial-$s$ derived conduction bands near the Fermi level, electrons on one Ni atom are transferred to the conduction bands under a reasonably large Hubbard interaction $U_{Ni}$ on Ni-$d$ orbitals, while electrons on the other two Ni atoms become more localized. This leads to a characteristic $Ni^{1+}$-$Ni^{2+}$-$Ni^{1+}$ CO stripe state. Then, we derive a low-energy effective model based on the realistic band structure of $NdNiO_2$. Using the effective model, we further elucidate that the $\mathbf{q} = (\frac{1}{3}, 0, 0)$ CO stripe state arises from a delicate competition between Hubbard interaction on Ni-$d$ orbitals and charge transfer energy between Ni-$d$ orbitals and conduction bands. Tuning the charge transfer energy controls not only the stability of the CO state but also its stripe pattern. With Hubbard interaction strength and charge transfer energy in a physically reasonable range, we find that the experimentally observed $\mathbf{q} = (\frac{1}{3}, 0, 0)$ stripe CO state is more stable than both the uniform paramagnetic (PM) state and the usual checkerboard antiferromagnetic state.

## Results

### DFT+DMFT calculations

Unlike cuprate superconductors, a consensus has not been reached regarding what is the minimal model to study infinite-layer nickelates. Therefore, we first downfold the DFT-calculated band structure of $NdNiO_2$ to a large 17-orbital tight-binding model by using the maximally localized Wannier functions (MLWFs)[45]. The model includes five Ni-$d$ orbitals, five Nd-$d$ orbitals, six O-$p$ orbitals and one interstitial-$s$ orbital per primitive cell. Our previous studies[25,46] show that this 17-orbital model is sufficiently large that it can almost exactly reproduce the non-interacting band structure of $NdNiO_2$ in an energy window of

about 15 eV around the Fermi level (see Supplementary Fig. 1). A Slater–Kanamori interaction is added on Ni-$d$ orbitals. We use DMFT to solve the interacting model. To test a possible $\mathbf{q} = (\frac{1}{3}, 0, 0)$ CO state, we use a $3 \times 1 \times 1$ supercell. Since in the 17-orbital model, the MLWF for the Ni-$d$ orbitals are very localized and atomic-like (see Supplementary Table 1), we use a Hubbard $U_{Ni} = 10$ eV and a Hund's exchange $J_{Ni} = 1$ eV, which has proved to be reasonable for this large energy window treatment of transition metal oxides[26,47–49]. More computational details are found in the Methods.

We first add perturbations on the three Ni atoms in the simulation cell. After the self-consistency loop is converged, we do find a CO state whose spectral function is shown in the Fig. 1a, b. For comparison, we also enforce symmetry on the three Ni atoms and thus obtain a uniform PM state whose spectral function is shown in Fig. 1c, d. Figure 1a, c shows the Ni-$d$ and O-$p$ projected spectral functions. In the CO state, we find two different types of Ni atoms. The spectral functions for Ni1-$d$ and Ni3-$d$ orbitals are identical (collectively referred to as Ni1-$d$). Ni2-$d$ orbital is of the other type. In the PM state, all three Ni atoms are equivalent due to the imposed symmetry. The inset shows the Ni-$d$ occupancy. We find that compared to Ni-$d$ occupancy in the uniform PM state, in the CO state, the Ni1-$d$ occupancy is larger and approaches $9e$, but the Ni2-$d$ occupancy is substantially smaller and is close to $8e$. This leads to a characteristic $Ni^{1+}$-$Ni^{2+}$-$Ni^{1+}$ stripe pattern. More importantly, the average Ni-$d$ occupancy in the CO state is smaller than that in the PM state, indicating that the CO is not a simple charge modulation between the three Ni atoms, but rather there is a net charge transfer from the Ni2 atom to other atoms. Figure 1b, d shows the Nd-$d$ and interstitial-$s$ projected spectral functions, which are almost identical between the CO and the PM states. However, close to the Fermi level, we find appreciable differences that Nd-$d$ and interstitial-$s$ states are more occupied in the CO state than in the PM state. Therefore, the emergence of CO state in $NdNiO_2$ is accompanied by a charge transfer from Ni2 atom to Nd-$d$/interstitial-$s$ orbitals that form conduction bands close to the Fermi level. We will show below that it is precisely

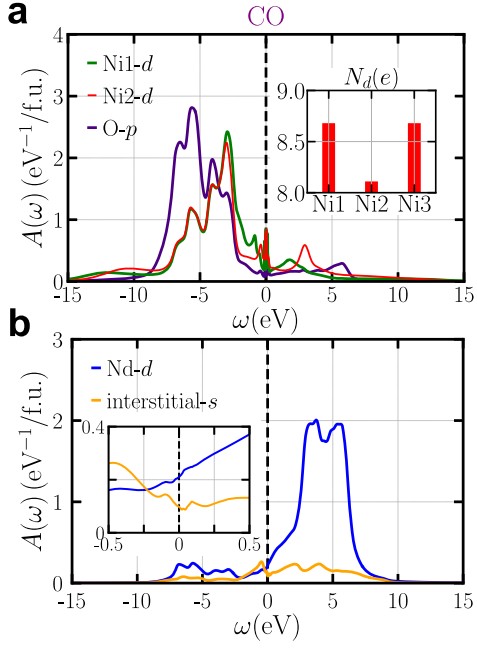
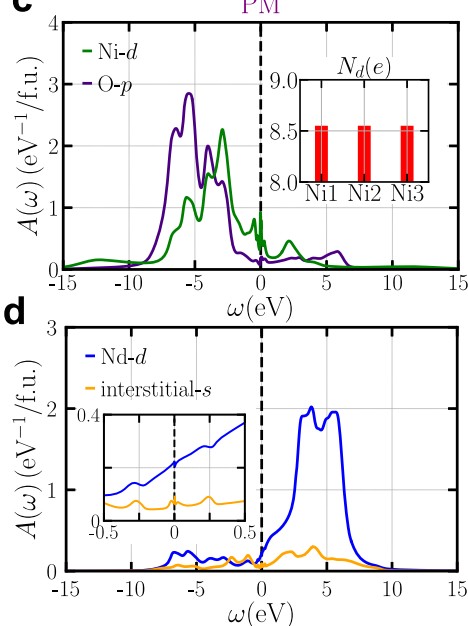
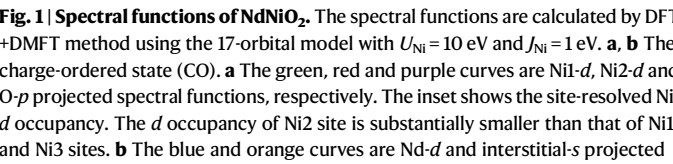

**Fig. 1 | Spectral functions of NdNiO₂.** The spectral functions are calculated by DFT+DMFT method using the 17-orbital model with $U_{Ni} = 10$ eV and $J_{Ni} = 1$ eV. **a, b** The charge-ordered state (CO). **a** The green, red and purple curves are Ni1-$d$, Ni2-$d$ and O-$p$ projected spectral functions, respectively. The inset shows the site-resolved Ni-$d$ occupancy. The $d$ occupancy of Ni2 site is substantially smaller than that of Ni1 and Ni3 sites. **b** The blue and orange curves are Nd-$d$ and interstitial-$s$ projected

spectral functions, respectively. The inset shows the energy range near the Fermi level. **c, d** The uniform paramagnetic state (PM). **c** The green and purple curves are Ni-$d$ and O-$p$ projected spectral functions, respectively. The inset shows the site-resolved Ni-$d$ occupancy. **d** identical to (**b**) but for the PM state. In all panels, the dashed line is the Fermi level, which is shifted to zero point. Source data are provided as a Source Data file.

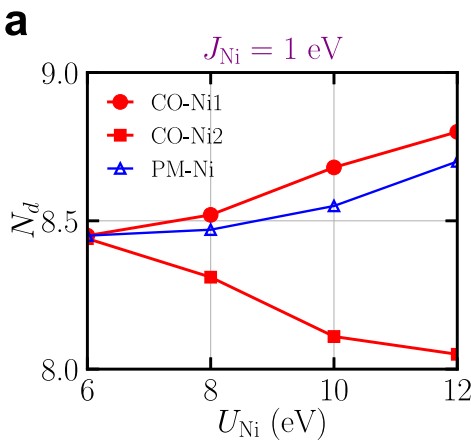

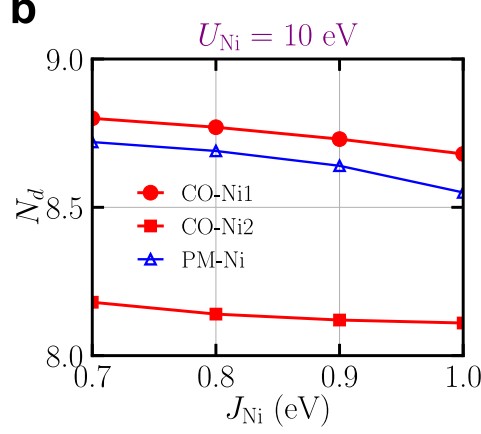

**Fig. 2 | Dependence of Ni-$d$ occupancy $N_d$ on $U_{Ni}$ and $J_{Ni}$.** The dependence of $N_d$ on $U_{Ni}$ and $J_{Ni}$ is calculated by using the 17-orbital model. **a** Ni-$d$ occupancy $N_d$ as a function of $U_{Ni}$ with $J_{Ni}$ = 1 eV. **b** Ni-$d$ occupancy $N_d$ as a function of $J_{Ni}$ with $U_{Ni}$ = 10 eV. In both panels, the red solid symbols and blue open symbols correspond to the Ni-$d$ occupancy in the charge-ordered (CO) and uniform paramagnetic (PM) states, respectively. Source data are provided as a Source Data file.

this charge transfer that can yield an energy gain and stabilize the CO state.

Next, we study how the charge disproportionation of the CO state depends on $U_{Ni}$ and $J_{Ni}$. Figure 2a shows the Ni-$d$ occupancy of both CO and PM states as a function of $U_{Ni}$ with $J_{Ni}$ being fixed at 1 eV. As $U_{Ni}$ exceeds 6 eV, the charge disproportionation of Ni-$d$ occupancy develops in the CO state and becomes more pronounced with $U_{Ni}$. On the other hand, changing $J_{Ni}$ in a reasonable range (from 0.7 to 1.0 eV) with $U_{Ni}$ being fixed at 10 eV does not strongly affect the charge disproportionation in the CO state. Furthermore, we also test the correlation effects from Nd-$d$ orbitals. By considering Hubbard interaction on Nd-$d$ orbitals, we do not find qualitative changes in the main results (see Supplementary Note 3).

## Low-energy effective model

To gain a deeper understanding of the origin and generality of the CO state that is found in the DFT+DMFT calculations of NdNiO₂, we build a low-energy effective model. We want to make the model simple, which aims to qualitatively reproduce the key features of the CO state. The advantage of such a simple effective model is that we can obtain an accurate self-energy so as to resolve the total energy difference between different competing phases. In addition, it also helps us identify the control parameters for the CO state. The effective model, equipped with proper energy dispersions, may also be applied to the study of the charge ordering phenomenon in other strongly correlated materials. From the preceding DFT+DMFT calculations, we find that there are two key ingredients in the formation of the CO state in NdNiO₂. One is the Hubbard interaction on Ni-$d$ orbitals and the other is the charge transfer from Ni-$d$ orbitals to the conduction bands that consist of Nd-$d$/interstitial-$s$ orbitals. In order to describe the interplay between these two factors, we include a correlated orbital and an auxiliary orbital in the effective model. To make a connection to infinite-layer nickelates, we choose Ni-$d_{x^2-y^2}$ orbital as the correlated orbital and add local Hubbard interaction on it. We also choose an effective-$s$ orbital as the auxiliary orbital that provides a free conduction band. We note that in real NdNiO₂, there are multiple conduction bands that are composed of Nd-$d$ and interstitial-$s$ orbitals. Here the effective-$s$ orbital (not to be confused with the interstitial-$s$ orbital) is employed to simplify the description of conduction bands. Similar models of correlated Ni-$d_{x^2-y^2}$ orbital plus free conduction bands have been used to study the low-energy physics of infinite-layer nickelates[17,25,32,50–53]. The full Hamiltonian of our effective model takes the form:

$$\hat{H} = \sum_{\mathbf{k}\sigma} \left[ \epsilon_d(\mathbf{k}) \hat{d}^\dagger_{\mathbf{k}\sigma} \hat{d}_{\mathbf{k}\sigma} + \epsilon_s(\mathbf{k}) \hat{s}^\dagger_{\mathbf{k}\sigma} \hat{s}_{\mathbf{k}\sigma} + \left( V_{ds}(\mathbf{k}) \hat{d}^\dagger_{\mathbf{k}\sigma} s_{\mathbf{k}\sigma} + \text{h.c.} \right) \right] + U_d \sum_i \hat{n}^d_{i\uparrow} \hat{n}^d_{i\downarrow} - \hat{V}^{dc} \quad (1)$$

where $\hat{d}^\dagger_{\mathbf{k}\sigma}$ ($\hat{s}^\dagger_{\mathbf{k}\sigma}$) is the creation operator on Ni-$d_{x^2-y^2}$ (effective-$s$ orbital) with momentum $\mathbf{k}$ and spin $\sigma$. $\hat{n}^d_{i\sigma} = \hat{d}^\dagger_{i\sigma} \hat{d}_{i\sigma}$ is the occupancy operator of Ni-$d_{x^2-y^2}$ orbital at site $i$ with spin $\sigma$. $\hat{V}^{dc}$ is the double counting potential[54]. The details of Eq. (1) are found in the Methods. In order to study the CO found in NdNiO₂, we fit all the hopping parameters in the energy dispersion and hybridization to the DFT band structure. The non-interacting part of Eq. (1) reproduces the near-Fermi-level DFT band structure of NdNiO₂, in which there are two bands crossing the Fermi level (see Supplementary Fig. 1). We mention one important parameter in the energy dispersion: the bare onsite energy difference between Ni-$d_{x^2-y^2}$ and effective-$s$ orbitals $E_{ds}$ = 0.70 eV (see the Methods), which we will show is another key control parameter for the CO state besides the Hubbard interaction on Ni-$d$ orbitals.

Next, we consider three competing states in the low-energy effective model (see Fig. 3a for a schematic): the uniform PM state with dynamically fluctuating spins (top), the checkerboard AFM (middle) and the CO state with an ordering wavevector $\mathbf{q} = (\frac{1}{3}, 0, 0)$ (bottom). Due to the simplicity of the model Eq. (1), we can obtain a highly accurate self-energy of Ni $d_{x^2-y^2}$ orbital, which enables us to calculate the total energy of each state as a function of $U_d$ and resolve the total energy difference between various competing phases (the details of the total energy calculations are found in the Methods). The results are shown in Fig. 3b. We find that when $U_d$ exceeds a critical value, the AFM and CO states can both be stabilized. More importantly, the CO state has lower energy than the AFM state and the PM state. To find a proper $U_d$ for our model, we calculate the effective mass of the Ni-$d_{x^2-y^2}$ orbital (see Supplementary Note 4 for details). We find that $U_d$ = 3.7 eV leads to an effective mass of 5.5, which exactly reproduces the result from the full ab initio GW+EDMFT calculations[55]. At this $U_d$ value, the CO state has lower energy than the PM state by about 50 meV/f.u., a magnitude reasonably consistent with the CO onset of infinite-layer nickelates in experiment[33–35]. Furthermore, the CO state is also more stable than the AFM state by about 30 meV/f.u. which may provide an explanation for the lacking of long-range AFM ordering in infinite-layer

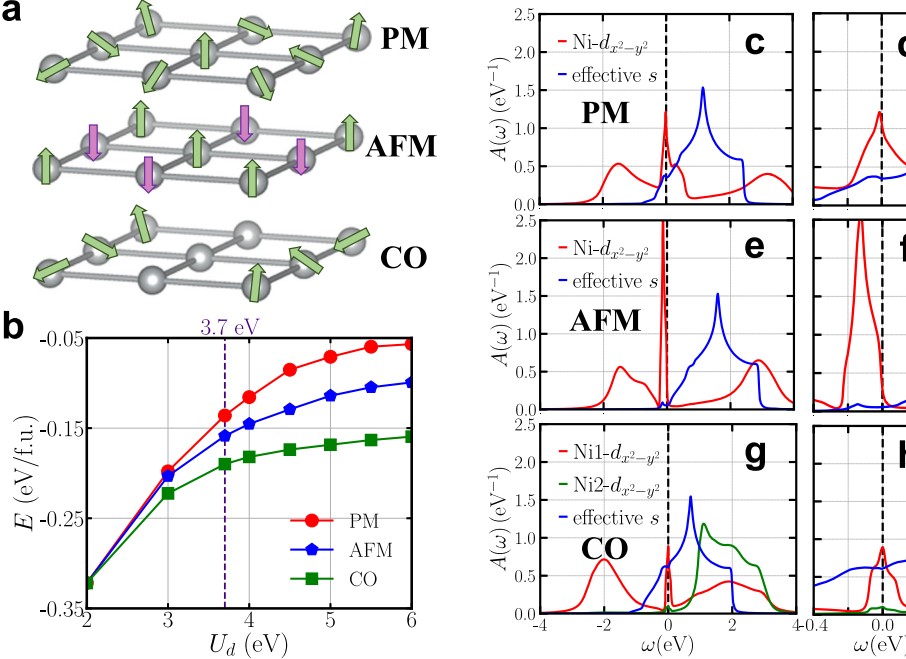

**Fig. 3 | Low-energy effective model. a** Schematics of the three competing states studied in the effective model: the uniform PM state (top), the checkerboard AFM state (middle) and the $\mathbf{q} = (\frac{1}{3}, 0, 0)$ CO state (bottom). **b** The total energy $E$ calculated using the low-energy effective model Eq. (1) (up to a constant) in the PM (red), the AFM (blue) and the CO (green) states as a function of $U_d$. The purple dashed line highlights $U_d = 3.7$ eV, which reproduces the Ni-$d_{x^2-y^2}$ orbital effective mass from the ab initio GW+EDMFT calculation[55]. The error bar is smaller than the symbol size (see Methods). **c**–**h** Orbital-resolved spectral functions calculated by using the low-energy effective model with $U_d = 3.7$ eV. **c** In the PM state. **e** In the AFM state. **g** In the CO state. **d**, **f** and **h** are the enlarged plots showing the corresponding near-Fermi-level features of the spectral function. Source data are provided as a Source Data file.

nickelates despite the substantial AFM superexchange found in experiment[14,15].

Figure 3c–h compares the spectral functions of the low-energy effective model Eq. (1) for the PM, AFM and CO states (calculated with $U_d = 3.7$ eV). Figure 3c displays the orbital-resolved spectral functions of the PM state. The red (blue) curve is the Ni-$d_{x^2-y^2}$ (effective-$s$) projected spectral function. Ni-$d_{x^2-y^2}$ orbital exhibits a characteristic three-peak feature: lower Hubbard band (LHB), upper Hubbard band (UHB) and a quasi-particle peak around the Fermi level. The effective-$s$ orbital has an occupancy of about 0.1$e$/f.u., due to the self-doping effect. This result is consistent with the previous studies using similar effective models[18,25,56]. Figure 3d zooms-in the Ni-$d_{x^2-y^2}$ projected spectral function close to the Fermi level, which highlights the quasi-particle peak and the partially occupied conduction band. Figure 3e, f shows the spectral function of the AFM state. Compared to the PM state, the quasi-particle peak is more pronounced but shifts away from the Fermi level. The self-doping effect is weaker in the AFM state. Figure 3g shows the orbital-resolved spectral functions of the CO state, where the red and green curves are for the two different types of Ni-$d_{x^2-y^2}$ orbital, and the blue curve is for the effective-$s$ orbital. Similar to the PM state, the Ni1-$d_{x^2-y^2}$ orbital in the CO state (red curve) exhibits a three-peak feature, but the Ni LHB moves toward lower energy. This indicates that the Ni1-$d_{x^2-y^2}$ orbital becomes more localized (i.e. closer to half-filling) in the CO state. In contrast, the Ni2-$d_{x^2-y^2}$ orbital in the CO state is almost empty. Compared to the PM state, the effective-$s$ orbital is more occupied. The spectral functions in Fig. 3g qualitatively reproduce the Ni$^{1+}$-Ni$^{2+}$-Ni$^{1+}$ stripe pattern and the charge transfer from Ni2 site to the conduction bands, which are found from the preceding DFT+DMFT calculations. Figure 3h shows an enlarged plot of the Ni-$d_{x^2-y^2}$ projected spectral functions near the Fermi level in the CO state. The Ni-$d_{x^2-y^2}$ quasi-particle peak in the CO state has a smaller peak value than in the PM state (calculated at the same $U_d$), indicating that in addition to local Hubbard interaction, the formation of long-range

CO further suppresses the quasi-particle peak (the effect is more pronounced at larger $U_d$, see Supplementary Note 6).

## Energetics

Next, we analyze the energetics of the low-energy effective model in more detail. In particular, we study why the CO state may have a lower energy than the AFM state. Figure 4a shows a schematic electronic structure of the effective model Eq. (1). The brown arrow indicates the charge transfer from a Ni site to neighboring effective-$s$ orbitals. The upper panel is the one before the charge transfer, while the lower panel is after the charge transfer (when $U_d$ exceeds the critical value). The CO can emerge if there is an overall energy gain from the charge transfer. However, the energy of Ni LHB is lower than the effective-$s$ orbitals. Thus, the electron transfer from the Ni2 site to the effective-$s$ orbitals seems to result in an energy cost. To solve this paradox, we compare the total energy of the CO and AFM states and decompose the energy difference into three contributions: $E = E^{\mathrm{kin}} + E^{\mathrm{pot}} - E^{\mathrm{dc}}$, where $E^{\mathrm{kin}}$ is the kinetic energy, $E^{\mathrm{pot}}$ is the potential energy and $E^{\mathrm{dc}}$ is the double-counting energy (see the Methods).

We first analyze the case of $U_d = 3.7$ eV, as highlighted by the purple dashed line in Fig. 4b. We find that compared to the AFM state, which does not have a charge transfer, the charge transfer indeed results in an energy cost in $E^{\mathrm{kin}}$ for the CO state. This is consistent with the schematics shown in Fig. 4a. However, the charge transfer also leads to an energy gain in $E^{\mathrm{pot}}$. That is because, in the AFM state, every Ni site has almost one electron on the $d_{x^2-y^2}$ orbital. Due to the repulsive Hubbard interaction, $E^{\mathrm{pot}} = \frac{1}{2}\mathrm{Tr}(\hat{\Sigma}_{\mathrm{cor}}\hat{G}_{\mathrm{cor}}) > 0$ for each Ni. In the CO state, the Ni1 and Ni3 sites have one electron on the $d_{x^2-y^2}$ orbital. But the charge transfer depletes the electron on the Ni2-$d_{x^2-y^2}$ orbital, so that its self-energy $\Sigma_{\mathrm{cor}}$ essentially becomes zero. This decreases the potential energy and always leads to an energy gain $\Delta E^{\mathrm{pot}} < 0$ when $U_d$ exceeds the critical value. As for the double-counting energy, $E^{\mathrm{dc}} = \frac{1}{2}U_d N_d(N_d - 1)$[57]. The AFM state is close to a half-filled configuration for every Ni site, while the CO state has a configuration in

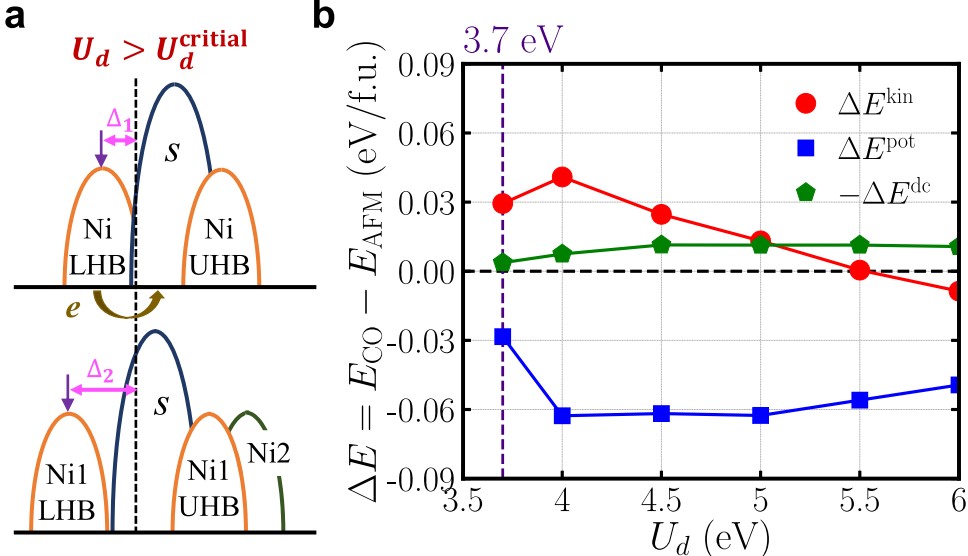

**Fig. 4 | Charge transfer and energetics. a** Schematics of the interacting electronic structure of the low-energy effective model Eq. (1). The brown arrow indicates the charge transfer from a Ni site to neighboring effective-*s* orbitals. The upper panel is the one before the charge transfer and the lower panel is after the charge transfer (when $U_d > U_d^{critical}$). **b** The total energy difference between the CO and the AFM states decomposed into kinetic energy (kin, red circles), potential energy (pot, blue squares) and double-counting energy (dc, green pentagons) as a function of $U_d$, calculated by using the low-energy effective model Eq. (1). The purple dashed line highlights $U_d = 3.7$ eV. The error bar is smaller than the symbol size (see Methods). Source data are provided as a Source Data file.

which Ni2 site has almost zero occupancy and Ni1/Ni3 site is half-filled. Therefore, in both states, the double counting energy $E^{dc}$ is small and so is their difference (compared to $\Delta E^{pot}$). In short, there is a delicate competition between charge transfer energy and local Hubbard interaction. When the potential energy gain outweighs the kinetic energy cost during the charge transfer, the CO state becomes more stable than the AFM state.

Then, we analyze the case of a general $U_d$ (see Fig. 4b). We find that the trend does not change in $\Delta E^{pot}$ and $\Delta E^{dc}$. However, when $U_d$ is sufficiently large, $\Delta E^{kin}$ itself becomes negative (i.e., a net energy gain after the charge transfer). That is because as $U_d$ increases, the Ni1/Ni3 LHB moves toward the lower energy after the charge transfer (i.e. $|\Delta_2| > |\Delta_1|$). This effect decreases the kinetic energy and eventually counteracts the kinetic energy cost from the charge transfer (brown arrow in Fig. 4a).

## CO stability and stripe pattern

The preceding analysis shows that the charge transfer from Ni-$d_{x^2-y^2}$ orbital to the near-Fermi-level conduction bands plays a crucial role in the formation of the CO state. In our effective model, the conduction bands are derived from the effective-*s* orbitals. Figure 5a is a schematic non-interacting electronic structure of the effective model Eq. (1). We show below that the charge transfer energy $E_{ds}$, which is defined as the energy separation between the bare onsite energy of Ni-$d_{x^2-y^2}$ and effective-*s* orbitals, controls not only stability but also stripe pattern of the CO state in the effective model.

We first examine the total energy of the PM, AFM and CO states as a function of $E_{ds}$ using the effective model Eq. (1). For each $E_{ds}$, we recalculate the total energy of each state and show the results in Fig. 5b. We use $U_d = 3.7$ eV but find a similar trend using other values of $U_d$ (see Supplementary Note 7). Fitting to the DFT band structure, we obtain $E_{ds} = 0.7$ eV (the orange dashed line in Fig. 5b), at which the CO state is more stable than the AFM and PM states. As $E_{ds}$ increases, the total energy of the CO state first becomes higher than the AFM state ($E_{ds} > 0.8$ eV) and then even higher than the PM state ($E_{ds} > 0.9$ eV). This can be understood because increasing $E_{ds}$ leads to a larger kinetic energy cost in the charge transfer (as we demonstrated in Fig. 4a).

Next, we study different stripe CO patterns and compare their total energies using the effective model Eq. (1). We use a $N \times 1 \times 1$ supercell to study the stripe CO with the wavevector $\mathbf{q} = (\frac{1}{N}, 0, 0)$. Before discussing the results, we first present an intuitive picture of how the CO total energy may depend on $N$. As we have shown, the CO state emerges from a charge transfer of almost one electron from one Ni site to the conduction bands for every $N$ Ni atoms. During the charge transfer process, the electron needs to overcome the charge transfer energy $E_{ds}$. This results in a kinetic energy cost $\propto E_{ds}$. On the other hand, the charge transfer depletes the charges on that Ni site. Since the local Hubbard interaction is repulsive in our model, this leads to a potential energy gain $\propto U_d$. Thus, the total energy of the entire system ($N$-Ni-atom cell) is:

$$E_{N-\text{Ni-atom}} = \alpha E_{ds} - \beta U_d + N E_0 \qquad (2)$$

where $\alpha$ and $\beta$ are both positive and $E_0$ is the energy contribution that does not change during the charge transfer. The tricky point is that both $\alpha$ and $\beta$ also depend on the wavevector or the wavelength (i.e., $N$). Therefore, the total energy per-Ni-atom cell is:

$$E_{\text{per-Ni-atom}} = \frac{E_{N-\text{Ni-atom}}}{N} = \alpha(N)\frac{E_{ds}}{N} - \beta(N)\frac{U_d}{N} + E_0 \qquad (3)$$

For given $E_{ds}$ and $U_d$, minimizing $E_{\text{per-Ni-atom}}$ with respect to $N$ yields the optimal wavelength $N$ or equivalently optimal wavevector $\mathbf{q}$. However, due to interaction, the analytical expressions of $\alpha(N)$ and $\beta(N)$ are not known. Therefore, we numerically calculate $E_{N\text{-Ni-atom}}$ as a function of $N$ for different $E_{ds}$ and $U_d$.

Figure 5c compares the total energy of different stripe CO states for a few representative $U_d$ at $E_{ds} = 0.7$ eV (DFT-fitted value). We find that at $E_{ds} = 0.7$ eV, the $\mathbf{q} = (\frac{1}{3}, 0, 0)$ CO has the lowest total energy in a range of $U_d$ values. The reason that the total energy of the stripe CO state does not monotonically change with $N$ is because with $N$ increasing, on the one hand, the kinetic energy cost decreases; on the other hand, the potential energy gain also decreases. These two factors are competing and the calculations find that the total energy minimum appears at an optimal CO wavevector $\mathbf{q} = (\frac{1}{3}, 0, 0)$. While our model is

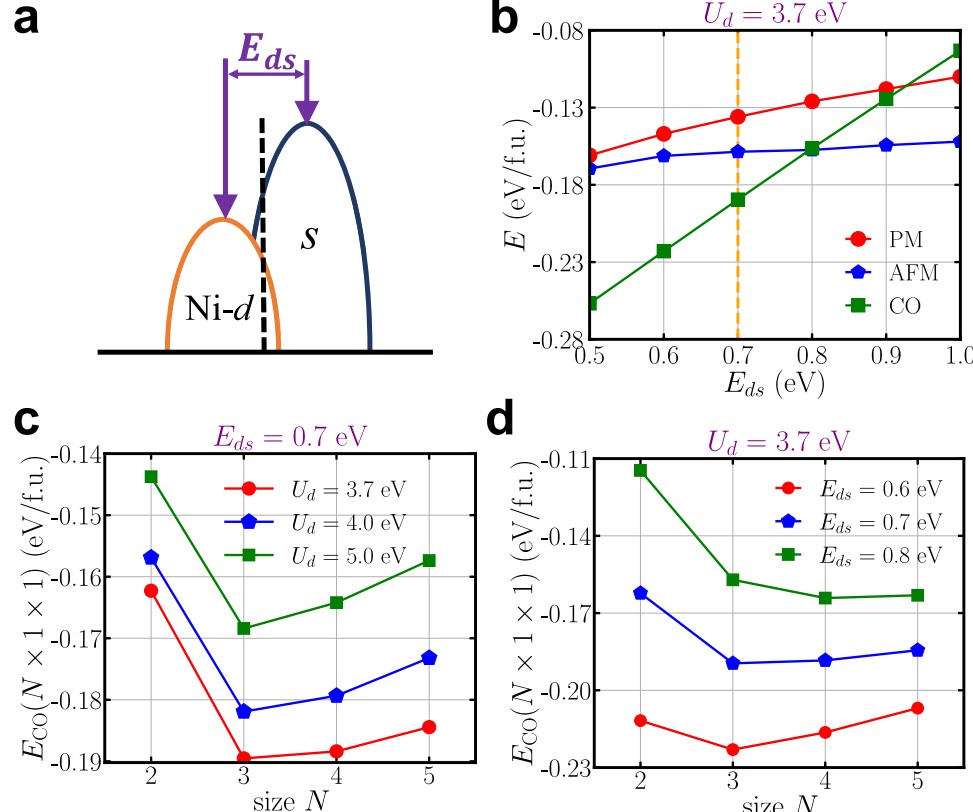

**Fig. 5 | CO stability and stripe pattern. a** Schematics of the non-interacting electronic structure of the low-energy effective model Eq. (1) where $E_{ds}$ is the energy separation between the bare energy of Ni-$d_{x^2-y^2}$ and effective-$s$ orbitals. **b** The total energy (up to a constant) of the PM (red), the AFM (blue) and the CO (green) states as a function of $E_{ds}$, calculated by using the low-energy effective model Eq. (1) with $U_d = 3.7$ eV. The orange dashed line highlights $E_{ds} = 0.7$ eV, which is obtained from fitting to the DFT band structure of NdNiO$_2$. **c** The total energy of different stripe CO states with the wavevector $\mathbf{q} = (\frac{1}{N}, 0, 0)$ as a function of $U_d$, calculated by using the low-energy effective model Eq. (1) with $E_{ds} = 0.7$ eV. **d** The total energy of different stripe CO with the wavevector $\mathbf{q} = (\frac{1}{N}, 0, 0)$ as a function of $E_{ds}$, calculated by using the low-energy effective model with $U_d = 3.7$ eV. In panels (**b**–**d**), the error bar is smaller than the symbol size (see Methods). Source data are provided as a Source Data file.

simplified, this result is consistent with the experimentally observed CO stripe pattern of infinite-layer nickelates[33–35]. Figure 5d compares the total energy of different stripe CO states for a few representative $E_{ds}$ at $U_d = 3.7$ eV. We find that as $E_{ds}$ increases from 0.6 to 0.8 eV, the wavevector of the lowest energy CO state evolves from $(\frac{1}{3}, 0, 0)$ to $(\frac{1}{4}, 0, 0)$. This is because as $E_{ds}$ changes, the decreasing rate of the kinetic energy with $N$ also changes, which leads to a different optimal wavevector for the CO state. We mention that when $E_{ds} \geq 0.8$ eV, the AFM state has lower energy than the CO state (see Fig. 5b). This means that based on our model calculations, the CO state with a wavevector of $\mathbf{q} = (\frac{1}{4}, 0, 0)$ cannot be observed. Instead, it gives way to the AFM state. To summarize, in the phase space spanned by $(U_d, E_{ds})$, our effective model Eq. (1) has a parameter range in which the stripe CO state is more stable than the AFM and PM states with an optimal wavevector $\mathbf{q} = (\frac{1}{3}, 0, 0)$, which reproduces the experimental observations of infinite-layer nickelates[33–35].

## Discussion

We make a few comments before the conclusion.

First, we note that CO or charge density wave (CDW) may have different origins. In addition to electron−electron interaction, electron−phonon interaction may also play an important role in the formation of CDW, which has been intensively studied in other material systems such as transition metal dichalcogenides[58]. A CDW state that arises from electron−phonon interaction is accompanied by a structural distortion. Thus, in addition to using the ideal lattice structure of infinite-layer nickelates, we also study possible structural

distortions in the CO state[59,60]. Since the charge transfer breaks the original translational symmetry, oxygen atoms may move away from their ideal positions (see Supplementary Note 5). The movement of oxygen atoms can be modeled by changing the onsite energy of Ni-$d_{x^2-y^2}$ orbital in the effective model Eq. (1). Our calculations find that the oxygen atom movements are minute (-0.05 Å), which is in agreement with the other study[61]. The resulting energy gain in CO is less than 5 meV/f.u., about one order of magnitude smaller than the energy difference shown in Fig. 3b. The energy gain may be used as a rough estimation to quantify the relative importance of different mechanisms. Our results imply that the CO state found in infinite-layer nickelates is more likely of electronic origin, while structural distortions driven by electron−phonon interaction may play a secondary role.

Second, the CO state from our calculations has a substantial charge disproportionation, driven by a charge transfer from a portion of Ni atoms to the near-Fermi-level conduction bands. A direct consequence is that the average Ni occupancy in the CO state is substantially smaller than that of a uniform PM state. Such a CO state is dissimilar to the CDW state found in the calculations of the doped single-band Hubbard model[62,63], doped $t$-$J$ model[64] and a 2-orbital model of ref. [53], in which the charge profile shows a Friedel oscillation with the average occupancy per site being equal to that of uniform PM state. In those states, variation in site occupancy is due to charge transfer between neighboring correlated sites.

Third, as we mentioned previously, the analytical expression of CO energy dependence on wavevector $E(\mathbf{q})$ is not known. Therefore, we cannot find the optimal wavevector $\mathbf{q}$ by solving $\nabla_{\mathbf{q}} E(\mathbf{q}) = 0$.

Instead, we have to test as many known competing states as possible (within computational capability). In addition to the $\mathbf{q} = (\frac{1}{N}, 0, 0)$ CO state, we also test a different CO state, which can be accommodated by the available simulation cell (see Supplementary Note 8). We use the low-energy effective model to study all these competing states (PM state, AFM state and five CO states with different wavevectors) and compare their total energy. Our calculations find that using the DFT-fitted energy dispersion for a reasonable range of $U_d$, the $\mathbf{q} = (\frac{1}{3}, 0, 0)$ CO state is the most energetically favorable one among all the states considered.

Fourth, a number of studies[27,31,65,66] have shown that Ni $d_{3z^2-r^2}$ orbital plays an important role in infinite-layer nickelates. Comparison of the low-energy effective model calculations to the 17-orbital model calculations, which include all five Ni-$d$ orbitals, reveals that the minimum physics needed to account for the CO phenomenon is a charge transfer from a correlated Ni-$d_{x^2-y^2}$ orbital to a conduction band that is close to the Fermi level. Adding other orbitals into the effective model may describe more physical phenomena of infinite-layer nickelates, but as far as charge ordering is concerned, the 2-orbital effective model is sufficient.

Finally, we briefly discuss how one may experimentally distinguish CO states of different origins in infinite-layer nickelates. If the CO in infinite-layer nickelates arises from the revealed charge-transfer mechanism, it will exhibit a mixed valence state of Ni, a characteristic $Ni^{1+}$-$Ni^{2+}$-$Ni^{1+}$ stripe pattern, as well as a unidirectional $Ni^{2+}$ chain in which holes are localized. In the experiment, one may probe this mixed valence state of Ni either by x-ray absorption spectroscopy (XAS) or cross-sectional electron energy loss spectroscopy (EELS)[36]. XAS can measure the average Ni valence, whose spectrum in the CO state should be distinct from that of pure $Ni^{1+}$ or $Ni^{2+}$. A more informative measurement is the cross-sectional EELS, which may provide a spatially resolved spectrum of $Ni^{1+}$ and $Ni^{2+}$ columns in the CO state. In addition, scanning tunneling microscopy can image the localized holes in the unidirectional $Ni^{2+}$ chain[36]. On the other hand, if the CO or CDW is driven by phonon and electron–phonon coupling, a characteristic phonon of the ideal crystal structure should become soft around the CDW transition temperature and the emergence of CDW accompanies a periodic-lattice-distortion, which lowers the total energy[58,67,68]. The vibrational mode of characteristic phonon should match the pattern of periodic-lattice-distortion observed at low temperatures[67,68]. The magnitude of the periodic-lattice-distortion should be such that the resulting energy gain is comparable to the CDW transition temperature[58]. In experiment, inelastic x-ray scattering or neutron scattering can measure temperature dependence of phonon dispersion and identify soft phonons, if they exist around CDW temperature[68]. The low-temperature periodic-lattice-distortion can be probed either by x-ray diffraction (XRD) or by transmission electron microscopy (TEM). The emergence of periodic-lattice-distortion lowers the crystal symmetry and thus changes the diffraction pattern in XRD measurements[69]. TEM, in particular with the recent developments of electron ptychography[70], can achieve atomic-resolution imaging, enabling visualization of periodic-lattice-distortion in real space and accurate measurements of structural distortions. Using the lattice distortions found from TEM measurements, one can estimate the energy gain from the periodic-lattice-distortion by performing first-principles calculations and then comparing the energy gain to the experimental CDW transition temperature. In short, the experimental observation of "a mixed valence state of Ni/localized holes on unidirectional $Ni^{2+}$ chain" versus "soft phonon/periodic-lattice-distortion" may distinguish different origins of CO in infinite-layer nickelates. We hope that the two sets of experiments outlined above, as well as other experimental techniques, will identify the true underlying mechanism among various theoretical proposals.

To conclude, our first-principles calculations show that a $\mathbf{q} = (\frac{1}{3}, 0, 0)$ CO state, which has a characteristic $Ni^{1+}$-$Ni^{2+}$-$Ni^{1+}$ stripe

pattern, is stabilized in infinite-layer nickelates when Hubbard interaction on Ni-$d$ orbitals is reasonably large. For every three Ni atoms, the CO state is driven by a charge transfer from one Ni atom to near-Fermi-level conduction bands, leaving electrons on the other two Ni atoms to become more localized. We further derive a simple low-energy effective model and elucidate that stabilization of the CO state over PM and AFM states is the consequence of a delicate competition between Hubbard interaction $U_d$ on a correlated orbital and charge transfer energy $E_{ds}$ between the correlated orbital and the conduction band. Such a competition may induce CO states in other correlated materials besides infinite-layer nickelates. Manipulating the CO state in infinite-layer nickelates by tuning the charge transfer energy may also affect their superconductivity[71], which deserves further experimental study.

## Methods

### DFT+DMFT calculations

We perform DFT+DMFT calculations[41–44] with the aid of MLWFs[45,72].

The DFT method is implemented in the Vienna ab initio simulation package code[73] with the projector augmented wave method[74]. The Perdew–Burke–Ernzerhof[75] functional is used as the exchange-correlation functional in DFT calculations. The Nd-$4f$ orbitals are treated as core states in the pseudopotential because the $4f$ shell is hidden in the core of the Nd ion, and Hubbard interaction on $4f$ orbitals is very large (>10 eV). As a result, the Nd-$4f$ electrons are far from the Fermi level and turn into strongly localized $f$ spins. The direct coupling between Nd-$4f$ electrons and itinerant electrons is minute[76]. We use an energy cutoff of 600 eV and sample the Brillouin zone by using $\Gamma$-centered $\mathbf{k}$-mesh of $12 \times 12 \times 12$ per primitive cell. The crystal structure is fully relaxed with an energy convergence criterion of $10^{-6}$ eV, force convergence criterion of 0.01 eV/Å and strain convergence of 0.1 kbar. The DFT-optimized crystal structures are in good agreement with the experimental $P4/mmm$ structures ($a^{DFT}$ = 3.91 Å, $a^{EXP}$ = 3.92 Å, $c^{DFT}$ = 3.31 Å, $c^{EXP}$ = 3.28 Å). To describe the checkerboard antiferromagnetic ordering, we expand the cell to a $\sqrt{2} \times \sqrt{2} \times 1$ supercell. To describe the $\mathbf{q} = (\frac{1}{N}, 0, 0)$ CO state ($N$ is an integer), we expand the cell to an $N \times 1 \times 1$ supercell.

We downfold the DFT-calculated band structure of NdNiO$_2$ to a 17-orbital tight-binding model by using the MLWF method[45]. The 17 orbitals include five Ni-$d$ orbitals, five Nd-$d$ orbitals, six O-$p$ orbitals and one interstitial $s$ orbital. This 17-orbital model can well reproduce the non-interacting band structure of NdNiO$_2$ in an energy window of about 15 eV around the Fermi level (see Supplementary Fig. 1). A Slater–Kanamori interaction is added on the Ni-$d$ orbitals[77]. A fully localized-limit (FLL) double counting correction is applied to the full interacting model[57]. We also test other double counting correction[48] and no qualitative change is found in the main results. More details of the full Hamiltonian are found in Supplementary Note 1. We treat the two Ni $e_g$ orbitals with DMFT method, while the filled Ni $t_{2g}$ shell is treated with a static Hartree–Fock approximation[56,78].

When we use the DMFT method to solve the interacting 17-orbital model, we employ the continuous-time quantum Monte Carlo (CTQMC) algorithm based on hybridization expansion[79,80]. The impurity solver was developed by Haule[81]. For each DMFT iteration, a total of 1 billion Monte Carlo samples are collected to converge the impurity Green function and self-energy. We set the temperature to be 116 K. We check all the main results at a lower temperature of 58 K and no significant difference is found. For the CO state, we allow all the Ni sites to be inequivalent and the DMFT self-consistent condition involves the self-energies of all inequivalent Ni atoms.

To obtain the spectral functions, the imaginary axis self-energy is continued to the real axis by using the maximum entropy method[82]. Then, the real axis local Green function is calculated using the Dyson equation. A $90 \times 90 \times 90$ $\mathbf{k}$-point mesh is used to converge the spectral function of the 17-orbital model.

**Table 1 | The onsite energy and hopping matrix elements in the 2-orbital model**

| in $\epsilon_d(\mathbf{k})$ | $t_d^{100} = -0.370$ | $t_d^{110} = 0.060$ | $t_d^{200} = -0.020$ |
|---|---|---|---|
| in $\epsilon_s(\mathbf{k})$ | $E_{ds} = 0.70$ | $t_s^{100} = 0.018$ | $t_s^{110} = -0.049$ |
| | $t_s^{101} = -0.191$ | $t_s^{111} = 0.075$ | $t_s^{001} = -0.237$ |
| in $V_{ds}(\mathbf{k})$ | $t_{ds}^{100} = -0.052$ | | |

All the units are eV.

The full charge-self-consistency does not qualitatively change the main results of charge transfer and charge disproportionation in the calculations[83]. To relieve the computational burden, the results shown in the main text are from one-shot DFT+DMFT calculations.

We also apply the above relevant parameters when using DMFT to solve the low-energy effective model.

### Low-energy effective model

The low-energy effective model to describe the CO phenomenon is a 2-orbital model, which can be compactly written as follows:

$$\hat{H} = \sum_{\mathbf{k}\sigma} \hat{\Psi}_{\mathbf{k}\sigma}^{\dagger} \mathcal{H}_0(\mathbf{k}) \hat{\Psi}_{\mathbf{k}\sigma} + U_d \sum_i \hat{n}_{i\uparrow}^d \hat{n}_{i\downarrow}^d - \hat{V}^{dc} \tag{4}$$

where $\hat{\Psi}_{\mathbf{k}\sigma}^{\dagger} = (\hat{d}_{\mathbf{k}\sigma}^{\dagger}, \hat{s}_{\mathbf{k}\sigma}^{\dagger})$ are the creation operators on Ni-$d_{x^2-y^2}$ and effective-$s$ orbitals with momentum $\mathbf{k}$ and spin $\sigma$. $\hat{n}_{i\sigma}^d = \hat{d}_{i\sigma}^{\dagger} \hat{d}_{i\sigma}$ is the occupancy operator of Ni-$d_{x^2-y^2}$ orbital at site $i$ with spin $\sigma$. $\mathcal{H}_0(\mathbf{k})$ is a $2 \times 2$ matrix:

$$\mathcal{H}_0(\mathbf{k}) = \begin{bmatrix} \epsilon_d(\mathbf{k}) & V_{ds}(\mathbf{k}) \\ V_{ds}^*(\mathbf{k}) & \epsilon_s(\mathbf{k}) \end{bmatrix} \tag{5}$$

The energy dispersion and hybridization terms are:

$$\epsilon_d(\mathbf{k}) = 2t_d^{100}\left(\cos k_x + \cos k_y\right) + 4t_d^{110}\cos k_x \cos k_y \\ + 2t_d^{200}\left(\cos 2k_x + \cos 2k_y\right) \tag{6}$$

$$\epsilon_s(\mathbf{k}) = E_{ds} + 2t_s^{100}\left(\cos k_x + \cos k_y\right) + 2t_s^{001}\cos k_z + 4t_s^{110}\cos k_x \cos k_y \\ + 4t_s^{101}\left(\cos k_x + \cos k_y\right)\cos k_z + 8t_s^{111}\cos k_x \cos k_y \cos k_z \tag{7}$$

$$V_{ds}(\mathbf{k}) = 2t_{ds}^{100}\left(\cos k_x - \cos k_y\right)\left(1 + e^{-ik_z}\right) \tag{8}$$

The onsite energy difference $E_{ds}$ and hopping parameters are obtained by fitting to the near-Fermi-level DFT band structure of NdNiO$_2$ (see Supplementary Fig. 1). The fitted values are shown in Table 1. We use the FLL double counting and $V^{dc}$ reads:

$$V^{dc} = U_d\left(N_d - \frac{1}{2}\right) \tag{9}$$

### Total energy

For the low-energy effective model, we calculate the total energy within the DFT+DMFT method. The total energy in the DFT+DMFT calculations has the following expression[44,84]:

$$E^{DFT+DMFT} = E^{DFT} + E^{DMFT} - E^{dc} \tag{10}$$

Here $E^{DFT}$ is:

$$E^{DFT} = E_0^{DFT} - E_0^{kin} \tag{11}$$

where $E_0^{DFT}$ is the standard DFT total energy. $E_0^{kin}$ is the kinetic energy of the non-interaction Hamiltonian within the model space:

$$E_0^{kin} = \text{Tr}\left(\hat{\mathcal{H}}_0 \hat{G}_0\right) = \frac{1}{N_{\mathbf{k}}} \sum_{\mathbf{k}\sigma} \sum_l \epsilon_{\mathbf{k}l} n_F(\epsilon_{\mathbf{k}l} - \mu) \tag{12}$$

where $n_F$ is the Fermi-Dirac occupancy, $l$ is the band index and $\sigma$ is the spin.

In the main text, we use the DFT-optimized crystal structure for all the states. Thus, the PM, AFM and CO states all share the same non-interacting Hamiltonian (up to a multiplication factor), we omit the $E^{DFT}$ in the calculation of total energy (since it is identical for all three states). The remaining total energy is

$$E = E^{DMFT} - E^{dc} = E^{kin} + E^{pot} - E^{dc} \tag{13}$$

Here, the DMFT kinetic energy $E^{kin}$ is defined as follows:

$$E^{kin} = \text{Tr}\left(\hat{\mathcal{H}}_0 \hat{G}\right) = \frac{T}{N_{\mathbf{k}}} \sum_{\mathbf{k}\sigma} \sum_{\omega_n} \sum_{mm'} \left[\mathcal{H}_0(\mathbf{k})\right]_{mm'} \left[G_\sigma(\mathbf{k}, i\omega_n)\right]_{m'm} \tag{14}$$

where $\sigma$ is the spin, $T$ is the temperature, $m, m'$ are the indices of orbital basis. $\mathcal{H}_0(\mathbf{k})$ is the non-interacting part of the effective model Eq. (5) and $G$ is the dressed lattice Green function of the interacting model. The DMFT potential energy $E^{pot}$ is defined as follows:

$$E^{pot} = \frac{1}{2}\text{Tr}\left(\hat{\Sigma}_{cor}\hat{G}_{cor}\right) = \frac{1}{2}T\sum_\sigma \sum_{\omega_n} \sum_m e^{-i\omega_n 0^-} \\ \left[\Sigma_{cor}^\sigma(i\omega_n)\right]_m \left[G_{cor}^\sigma(i\omega_n)\right]_m \tag{15}$$

where $\sigma$ is the spin, $T$ is the temperature, $m$ is the index of orbital basis. $\Sigma_{cor}$ and $G_{cor}$ are the self-energy and the local Green function in the correlated subspace. $E^{dc}$ is the double counting energy. We employ the FLL double counting and $E^{dc}$ reads:

$$E^{dc} = \frac{1}{2}U_d N_d(N_d - 1) \tag{16}$$

Taking the derivative of $E^{dc}$ with respect to $N_d$ yields the double-counting potential $V^{dc}$ Eq. (9). In the low-energy effective model, the Hund's term vanishes because there is only one correlated orbital Ni-$d_{x^2-y^2}$ per site.

We note that when we study the structural distortions in the CO state, the crystal structure changes in each calculation and so does the DFT total energy. In that case, we need to use Eq. (10) to calculate the DFT+DMFT total energy.

Since the total energy is calculated using CTQMC, we average the total energy of the last ten DMFT iterations. We find that due to the highly accurate self-energy, the standard deviation of the total energy is about 1 meV/f.u., which is smaller than the symbol size in the figures.

### Data availability

The data that support the findings of this study are available from the corresponding author upon reasonable request. Source Data are provided with this paper.

### Code availability

The electronic structure calculations were performed using the proprietary code VASP[73], the open-source code Wannier90[72] and the open-source impurity solver implemented by Kristjan Haule at Rutgers University (http://hauleweb.rutgers.edu/tutorials/). Both Wannier90 and Haule's impurity solver are freely distributed on academic use under the Massachusetts Institute of Technology (MIT) License.

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

## Acknowledgements

We thank Andrew Millis, Fu-Chun Zhang, Wei-Sheng Lee, Hyowon Park and Pu Yu for their stimulating discussions. H.C. is supported by the National Key R&D Program of China (No. 2021YFE0107900) and Science and Technology Commission of Shanghai Municipality under grant number 23ZR1445400. This work is also supported by the National Natural Science Foundation of China (Grant No. 11774236, No. 11974397), the Ministry of Science and Technology of China (No. 2017YFA0302902, No. 2017YFA0303103) and the Strategic Priority Research Program of the Chinese Academy of Sciences (Grant No. XDB33010100). High-Performance-Computing of New York University (NYU-HPC) provides the computational resources.

## Author contributions

H.C., Y.-f.Y. and G.-M.Z. conceived the project. H.C. and H.L. constructed the effective model and performed the calculations. H.C., Y.Y. and G.Z. wrote the manuscript. All the authors participated in the discussion.

## Competing interests
The authors declare no competing interests.
