## [Peer Review File · Nature Communications]

REVIEWER COMMENTS

Reviewer #1 (Remarks to the Author):

The authors investigated the origin of the charge ordering in NdNiO₂, using the DMFT approach. They used a 3*1*1 supercell that represents the experimentally observed Q vector.

The referee has two questionable points.

If the referee understand correctly, they investigated the undoped NdNiO₂ compound. If so, how does the Ni¹⁺ - Ni²⁺ - Ni¹⁺ ordering appear? The system has formally Ni¹⁺ configuration.

Secondly, the authors focused on the single Ni d(x²-y²) orbital in the low energy effective model. However, as well known now, in the system, the single band picture is not proper. Instead of this orbital, Ni d(z²) also plays an important role.

Reviewer #2 (Remarks to the Author):

In this paper, Chen et al. perform a DFT+DMFT analysis of the infinite-layer nickelates motivated by the charge-density-waves previously reported experimentally. Specifically, they map the DFT electronic structure to a 17-orbital tight-binding model to then treat the correlations within DMFT. In addition, they also consider a simplified low-energy model.

The work is timely and sound from the technical point of view. However, I think that the paper remains rather technical since the actual physics behind the reported results is not discussed in sufficient detail. In particular,

- The modeling and the results largely base on the presence of interstitial-s states derived from the Wannierization and effective-s states used to simplify the conduction bands. The physics behind these states and their connection should be discussed in the text. In fact, the connection between the 17-orbital model and the simplified model should elaborated in more detail and in more physical terms. Otherwise, the added value of reporting on the simplified vs more complex model is rather unclear. What do we learn from such a double exercise?

- The connection with previous works should also be elaborated in more detail. Beyond the wavevector of the CDW modulation, are there other observables that can be compared with the experiments? What is the main difference between the present model and that studied by Peng et al. in arXiv:2110.07593?

- Is the present model able to reproduce the cuprate CDW limit? If so, what is new (or what is different) beyond the multiband aspect? Does the metal (nickelates) vs insulating (cuprates) behavior play a crucial role for the present findings?

- I wonder if the absolute minimum of the present models corresponds to $q = (1/3, 0, 0)$ or to a different wavevector. I know that to answer to this question via explicit calculations can be very expensive. However, I would like to see some physical discussion of this point at least. Does the multiband character of the nickelates explain that? Why $1/3$?

- Is it possible to quantify the relative importance of the pure electronic vs electron-phonon coupling mechanism for the CDW formation?

Reviewer #3 (Remarks to the Author):

This is a well written manuscript that merits publication somewhere. It proposes a mechanism to understand the recently discovered charge modulations in nickelates. Those charge density waves add to the growing body of evidence that there are many common aspects between copper and nickel based superconductors.

On the other hand, there are clear differences building up, such as the importance of two Ni orbitals in nickelates while one appears to be enough for the Cu in cuprates (not counting oxygen orbitals here).

The scenario proposed by the authors is intriguing. They arrive to a pattern $Ni^+ - Ni^{2+} - Ni^+$ compatible with experiments with regards to the dominant wave vector. They present a very nice simplified model in Eq.1 involving one Ni orbital x^2-y^2 and an effective s-orbital which plays the role of linking the localized and itinerant degrees of freedom.

However, I find three critiques to this paper.

Problem1 is that there are many other theory papers already published, or in preprint form, circulating as we speak that arrive to the same conclusions using other scenarios. Given the wavevector of the experiments, certainly theorists will find the way to get that result via various channels. Some other authors emphasize phonons for example and also find results matching experiments. How can we tell them apart? Not all of them can appear in Nat Comm.

Problem2 is on page 7, sentence "Next, we consider three competing states ...". This means that the authors focused on only three states. Considering that the PM state is unlikely to be stable in this region of parameters, essentially it is just a competition between two states. So, we do not know whether the true ground state of the effective model is the CO state that the authors are focusing on. It is just the best among three. Basically it is a variational calculation.

Problem3 is that I do not see clear fingerprint predictions of this state so that experiments can be used to distinguish among the many possibilities in the literature that have appeared in print or are about to appear in print.

As a consequence, I believe this paper is more appropriate for npj QM because it is introducing a nice theoretical proposal, but it is not a "close to final answer" to the origin of the CO state in the nickelates. It is simply a (very nice) proposal.

My recommendation is to publish in npj QM as it is (or maybe the authors can consider adding predictions for future experiments to verify their scenario).

Reply to Reviewers (NCOMMS-22-53047-T)

We would like to thank all the three reviewers for their helpful questions and comments. In the process of answering these questions/comments and revising the text, our manuscript has improved. Below, we address all their concerns in detail. The resulting modifications to the manuscript clarify some of the most important points of our work.

For each reviewer, we address each question/comment by first quoting the question/comment followed by our reply.

Response to Reviewer #1

1) If the referee understand correctly, they investigated the undoped NdNiO₂ compound. If so, how does the Ni₁⁺ - Ni₂⁺ - Ni₁⁺ ordering appear? The system has formally Ni₁⁺ configuration.

We thank the reviewer for this comment. In the uniform paramagnetic state of NdNiO₂, Ni has a nominally d^9 occupancy and therefore its valence is Ni¹⁺. In the charge ordered state, we find that for every three Ni atoms, almost one electron transfers from one of the Ni atoms (labelled as Ni₂) to the near-Fermi-level conduction bands, leaving Ni₂ in a valence state of 2+. This charge transfer is precisely the origin of $\mathbf{q} = (\frac{1}{3}, 0, 0)$ charge order.

In the revised version, we modify text on Page 5 (Lines 95-98 and 105-108) to make this point more clear.

2) The authors focused on the single Ni d(x²-y²) orbital in the low energy effective model. However, as well known now, in the system, the single band picture is not proper. Instead of this orbital, Ni d(z²) also plays an important role.

We thank the reviewer for this important comment. First we constructed a 17-orbital model which includes five Ni- d orbitals, five Nd- d orbitals, six O- p orbitals and one

FIG. R1: (a) The band structure computed from the non-interacting part of the 3-orbital effective model (red dots) is compared to the DFT-calculated band structure of NdNiO₂ (blue curves). The black dashed line highlights the Fermi level and the coordinates of the high-symmetry \mathbf{k} -points are: $\Gamma(0, 0, 0)$, $X(0.5, 0, 0)$, $M(0.5, 0.5, 0)$, $Z(0, 0, 0.5)$, $R(0.5, 0, 0.5)$, $A(0.5, 0.5, 0.5)$. (b) The total energy E calculated using the 3-orbital effective model in the PM (red), the AFM (blue) and the $\mathbf{q} = (\frac{1}{3}, 0, 0)$ CO (green) states as a function of U_d .

interstitial- s orbital. This 17-orbital model takes into account all the relevant orbital of NdNiO₂ and can almost exactly reproduce the band structure of NdNiO₂ in a large energy window of about 15 eV around the Fermi level (see Fig. 1 in the Supplementary Information). When adding local interactions on Ni- d orbitals (including Ni- $d_{3z^2-r^2}$ orbital), we find that a charge ordered (CO) state can be stabilized due to a charge transfer from Ni2 atom to conduction bands (see our reply to comment #1). Our low-energy effective model focuses on the key ingredients of the 17-orbital model and reproduces the charge-transfer-driven CO state. This shows that as far as charge ordering is concerned, the simplified 2-orbital model (a correlated Ni- $d_{x^2-y^2}$ orbital plus an effective s orbital) is sufficient to reveal the underlying physics.

Nevertheless, we appreciate the concern from the reviewer. Thus we build a new 3-orbital effective model which includes both Ni $d_{x^2-y^2}$ and $d_{3z^2-r^2}$ orbitals as well as an effective s orbital. Panel (a) of Fig. R1 compares the band structure computed from the non-interacting part of the 3-orbital effective model (red dots) to the DFT-calculated band structure of NdNiO₂ (blue curves). The band that lies in the energy window between -4

eV to -1 eV below the Fermi level is derived from Ni $d_{3z^2-r^2}$ orbital [1]. Then we add local interaction on Ni- d orbitals and calculate the phase diagram of this 3-orbital effective model as a function of U_d (shown in panel (b) of Fig. R1). We find that the phase diagram of the 3-orbital effective model is qualitatively similar to that obtained from the 2-orbital effective model (Fig. 3(b) in the main text). This confirms that both effective models (2-orbital and 3-orbital) include sufficient physics to reproduce the CO phenomenon found from the 17-orbital model calculations. In this study, we choose the 2-orbital model for its simplicity, which enables us to study CO of longer wavelength (i.e. larger simulation cell, see Fig. 5(c) and 5(d) in the main text). Of course, we agree with the reviewer that Ni $d_{3z^2-r^2}$ orbital plays an important role in other aspects of infinite-layer nickelates [2–4].

In the revised version, we add discussion about Ni- $d_{3z^2-r^2}$ orbital on Page 15 (Lines 321-328). In the upcoming work [5], we will make an extensive comparison between the 2-orbital model and the 3-orbital model for various physical properties of infinite-layer nickelates and present Fig. R1 along with other results. We appreciate the reviewer’s understanding.

Response to Reviewer #2

We thank the reviewer for the comment “The work is timely and sound from the technical point of view.”.

1) The modeling and the results largely base on the presence of interstitial-s states derived from the Wannierization and effective-s states used to simplify the conduction bands. The physics behind these states and their connection should be discussed in the text. In fact, the connection between the 17-orbital model and the simplified model should elaborated in more detail and in more physical terms. Otherwise, the added value of reporting on the simplified vs more complex model is rather unclear. What do we learn from such a double exercise?

We thank the reviewer for this insightful comment. The reasons that we used two models

to study the charge order in infinite-layer nickelates and the connection between the two models are as follows:

1) Unlike cuprate superconductors, consensus has not been reached regarding what is the minimal model to study infinite-layer nickelates [6]. In literature, various downfolded models have been used to study different aspects of infinite-layer nickelates, such as superconductivity and magnetism. However there are debates like whether oxygen p states can be ignored in the modelling or not [7]. Therefore when we study charge order in infinite-layer nickelates, we first used a large 17-orbital model. Such a model includes all the relevant orbitals of infinite-layer nickelate and it almost exactly reproduces the DFT band structure of NdNiO_2 in an energy window of about 15 eV around the Fermi level (see Fig. 1 in the Supplementary Information). We believe this 17-orbital model can faithfully describe the actual material NdNiO_2 and by adding local Hubbard interaction on Ni- d orbitals, we find a stabilized charge ordered state that is characterized by a $\text{Ni}^{1+}\text{-Ni}^{2+}\text{-Ni}^{1+}$ stripe pattern.

2) From the 17-orbital model calculations, we can extract key ingredients and reveal underlying mechanisms for the charge order in infinite-layer nickelates. We find that the key ingredients are a correlated orbital (here Ni- $d_{x^2-y^2}$ orbital) and near-Fermi-level conduction bands (here Nd- d orbitals and interstitial- s orbital). The underlying mechanism for the charge order is a charge transfer from the correlated orbital of one Ni site to the conduction bands, which leads to the $\text{Ni}^{1+}\text{-Ni}^{2+}\text{-Ni}^{1+}$ stripe pattern.

3) Based on the essential ingredients, we then construct a “minimal” effective model, which is sufficient to demonstrate this charge-transfer-mechanism and reproduce the $\text{Ni}^{1+}\text{-Ni}^{2+}\text{-Ni}^{1+}$ stripe pattern that is obtained from the 17-orbital model calculations. The effective model is built in its simplest possible form: it consists of only a correlated d orbital and a free conduction band which we use an effective s orbital to describe.

4) The “added value” of the effective 2-orbital model is its simplicity and generality. First we note that the transition temperature for the charge order in infinite-layer nickelates is about room temperature, which implies that the energy difference between the charge ordered state and the uniform paramagnetic state is on the order of a few tens of meV per formula. We used the continuous-time quantum Monte Carlo algorithm in the impurity

solver, which inevitably results in statistical errors in the calculation of total energy [6]. Precisely because of its simplicity, the self-energy of the effective 2-orbital model can be accurately determined, which successfully yields a statistical error of about 1 meV/f.u. in the total energy calculation. Such a small statistical error enables us to clearly resolve the total energy difference between various competing phases. In addition to its simplicity, the effective 2-orbital model is general in that equipped with a proper energy dispersion, it may be applicable to charge ordering phenomenon in other strongly correlated materials that also arises from the same charge-transfer mechanism.

In the revised version, we add and modify text on Pages 3 and 6 (Lines 73-76, 78-80 and 122-126) to elaborate this important point.

2) The connection with previous works should also be elaborated in more detail. Beyond the wavevector of the CDW modulation, are there other observables that can be compared with the experiments? What is the main difference between the present model and that studied by Peng et al. in arXiv:2110.07593?

We thank the review for this important comment, in particular, about the difference between our work and arXiv:2110.07593.

Besides the charge order wavevector, another important observable is the mixed valence state of Ni. From our calculations, the charge order of infinite-layer nickelates has a characteristic $\text{Ni}^{1+}\text{-Ni}^{2+}\text{-Ni}^{1+}$ stripe pattern. In experiment, one may probe this mixed valence state of Ni either by x-ray absorption spectroscopy (XAS) or cross-sectional electron energy loss spectroscopy (EELS) [9]. XAS measures the average Ni valence, whose spectrum in the charge ordered state should be distinct from that of pure Ni^{1+} or Ni^{2+} . Cross-sectional EELS can provide a spatially resolved spectrum of Ni^{1+} and Ni^{2+} columns in the charge ordered state. Furthermore, our calculations show that in the charge ordered state of infinite-layer nickelates, holes are localized in the unidirectional $\text{Ni}2$ chain (see more discussion below). This may be imaged by scanning tunneling microscopy (STM) [9].

We thank the reviewer for bringing our attention to arXiv:2110.07593, which was already cited in our paper (now it is Ref. [52] in the revised version). The authors of

FIG. R2: Occupancy profile in real space $n(x)$ calculated using different models. x is the lattice spacing. (a) Our low-energy effective model. The occupancy profile shows a charge order with a wavevector $\mathbf{q} = (\frac{1}{3}, 0, 0)$. (b) 2-orbital model from arXiv:2110.07593. The occupancy profile shows a charge order with a wavevector $\mathbf{q} = (\frac{3}{16}, 0, 0)$. (c) Single-orbital Hubbard model on a square lattice with $U/t = 12$ and $t'/t = 0.05$. The hole doping $\delta = \frac{1}{8}$. The occupancy profile shows a charge order with a wavevector $\mathbf{q} = (\frac{1}{8}, 0, 0)$. (d) Single-orbital $t - t' - J$ model on a square lattice with $J/t = \frac{1}{3}$ and $t'/t = \frac{1}{12}$. The hole doping $\delta = \frac{1}{8}$. The occupancy profile shows a charge order with a wavevector $\mathbf{q} = (\frac{1}{8}, 0, 0)$. Panels (c) and (d) are adapted from [8].

arXiv:2110.07593 also studied a 2-orbital model which consists of a correlated Ni $d_{x^2-y^2}$ orbital and a rare-earth d orbital. The key difference between our work and arXiv:2110.07593 is the nature of charge order. For ease of comparison, Fig. R2 shows our results (panel (a)) as well as the Ni charge modulation found in arXiv:2110.07593 (panel (b)). In our 2-orbital model calculations, both Ni1 and Ni3 sites have an occupancy of 0.99, while Ni2 site has an occupancy of 0.03 in the charge ordered state. The total occupancy of the

conduction bands (i.e. the effective s orbital) is close to 1. This is a clear evidence of charge transfer from Ni2 site to the conduction bands. By contrast, the 2-orbital model calculation in arXiv:2110.07593 found a very different charge modulation on Ni sites: Ni charge profile shows a Friedel oscillation with the average Ni occupancy being around 0.9 and a small variation of about ± 0.03 . The corresponding wavevector is $\mathbf{q} = (\frac{3}{16}, 0, 0)$. This indicates that the charge density wave found in arXiv:2110.07593 is due to charge transfer *between Ni sites* rather than between one Ni site and conduction bands. In order to see this point more clearly, we also show that the uniform paramagnetic state from our 2-orbital model has a Ni occupancy of 0.89 per site (the reason that it deviates from half-filling is due to the self-doping [10]). That value is close to the average Ni occupancy found in arXiv:2110.07593. By contrast, the average Ni occupancy of the charge ordered state found in our 2-orbital model calculation is $(0.99 + 0.03 + 0.99)/3 = 0.67$, which is substantially smaller. In addition, we also show the charge modulation of single-orbital Hubbard model and single-orbital t - t' - J model both with $\delta = \frac{1}{8}$ hole doping in panels (c) and (d) of Fig. R2 [8]. Those models do not have conduction bands. However, the charge density wave obtained in those model calculations is qualitatively similar to that found in arXiv:2110.07593. The charge variation is about ± 0.05 per site and more importantly the average occupancy of correlated orbital is very close to that in a uniform paramagnetic state.

The reason that our results of charge order are different from arXiv:2110.07593 is probably because we use different tight-binding band structure in the model calculations. The tight-binding band structure of our 2-orbital model is three-dimensional (3D) and satisfactorily reproduces the two bands of NdNiO₂ that cross the Fermi level in the DFT calculations (see panel (a) of Fig. R3). In arXiv:2110.07593, the authors modify the 3D dispersive Nd d_{z^2} band into a 2D dispersive one so that they can solve the interacting model using the density-matrix renormalization group (DMRG) method. The comparison between our work and arXiv:2110.07593 implies that the 3D dispersive nature of conduction bands may play a crucial role in the charge transfer from Ni- d orbitals to conduction bands as well as the resulting charge ordered state in infinite-layer nickelates.

In the revised version, we add and modify text on Pages 14 and 15 (Lines 304-310) to explain this important point.

FIG. R3: (a) The band structure computed from the non-interacting part of the 2-orbital effective model (red dots) is compared to the DFT-calculated band structure of NdNiO_2 (blue curves). (b) The band structure computed from the non-interacting part of the 2-orbital effective model (red dots) is compared to the DFT-calculated band structure of CaCuO_2 (blue curves). In both panels, the black dashed line highlights the Fermi level and the coordinates of the high-symmetry \mathbf{k} -points are: $\Gamma(0, 0, 0)$, $X(0.5, 0, 0)$, $M(0.5, 0.5, 0)$, $Z(0, 0, 0.5)$, $R(0.5, 0, 0.5)$, $A(0.5, 0.5, 0.5)$.

3) Is the present model able to reproduce the cuprate CDW limit? If so, what is new (or what is different) beyond the multiband aspect? Does the metal (nickelates) vs insulating (cuprates) behavior play a crucial role for the present findings?

We thank the reviewer for this comment.

The parent compounds of cuprate superconductors are antiferromagnetic insulators. In the underdoped region, they also exhibit charge ordering (CO) or charge density wave (CDW) phenomenon [11, 12]. The wavevector of CDW in cuprates is smaller than that of infinite-layer nickelates (i.e. CDW in cuprates has a longer wavelength) [9]. However, a more important difference is that in cuprates, the charge modulation arises from charge transfer between neighboring Cu sites (more precisely speaking, it is charge transfer between $O-p$ orbitals that are hybridized with $\text{Cu-}d_{x^2-y^2}$ orbital because holes reside on oxygen atoms in cuprates) [9]. There are no conduction bands involved in the cuprate

CDW state because conduction bands of cuprates (such as those derived from Sr- d /Ca- d orbitals) are a few eV above the Fermi level. In infinite-layer nickelates, as our calculations show, the charge modulation is due to charge transfer from a portion of Ni sites to neighboring conduction bands that are close to the Fermi level.

If we understand the question correctly, the reviewer asks whether our low-energy effective model may also reproduce the CDW state observed in underdoped cuprates? Our answer is that if the model uses an energy dispersion fitted to DFT band structure of infinite-layer nickelates (e.g. NdNiO₂, see Fig. R3(a)), then our calculations find a CO state whose nature is distinct from that in cuprates, as we explained above. If the model uses an energy dispersion fitted to DFT band structure of a cuprate (e.g. CaCuO₂, see Fig. R3(b)), then the key parameter E_{ds} will be much larger because Ca- d bands are a few eV higher in energy than Cu- $d_{x^2-y^2}$ band. We find that E_{ds} is 0.7 eV for NdNiO₂ and 3.5 eV for CaCuO₂. Such a large E_{ds} from the fitting of CaCuO₂ band structure would suppress the “self-doping” effect that is observed in infinite-layer nickelates [13] and thus the conduction band, albeit present in the model, does not participate in low-energy physics. Then the 2-orbital effective model is essentially reduced to a single-orbital Hubbard model. In the underdoped region, single-orbital Hubbard model calculations do find a one-dimensional Friedel oscillation in the charge profile [8, 14].

We note that the metal (nickelates) vs insulating (cuprates) behavior does play a crucial role for the present findings in that the metallicity of infinite-layer nickelates is precisely due to the presence of near-Fermi-level conduction bands. Those conduction bands are essential in the charge transfer process, which leads to the CO state and the characteristic Ni¹⁺-Ni²⁺-Ni¹⁺ stripe pattern in infinite-layer nickelates.

If we misunderstand the reviewer’s question, we kindly request the reviewer to clarify it and we are happy to address it again.

4) I wonder if the absolute minimum of the present models corresponds to $\mathbf{q} = (1/3, 0, 0)$ or to a different wavevector. I know that to answer to this question via explicit calculations can be very expensive. However, I would like to see some physical discussion of this point at least. Does the multiband character of the nickelates explain that? Why 1/3?

We thank the reviewer for this important comment. The multi-band character of infinite-layer nickelates (in particular, the presence of conduction bands near the Fermi level) plays a crucial role. Here we provide some intuitive discussion: consider a charge ordered (CO) state with a wavevector $\mathbf{q} = (\frac{1}{N}, 0, 0)$. From our calculations, we find that starting from a uniform paramagnetic state, such a CO state emerges from a charge transfer of almost one electron from one Ni site to the conduction bands for every N Ni atoms. During the charge transfer process, the electron needs to overcome the charge transfer energy E_{ds} . This results in a kinetic energy cost $\propto E_{ds}$. On the other hand, the charge transfer depletes the charges on that Ni site. Since the local Hubbard interaction is repulsive in our model, this leads to a potential energy gain $\propto U_d$. Thus the total energy of the entire system (N -Ni-atom cell) is:

$$E_{N\text{-Ni-atom}} = \alpha E_{ds} - \beta U_d + N E_0 \quad (\text{R1})$$

where α and β are both positive and E_0 is the energy contribution that does not change during the charge transfer. The tricky point is that both α and β also depend on the wavevector or the wavelength (i.e. N). Therefore the total energy per-Ni-atom cell is:

$$E_{\text{per-Ni-atom}} = \frac{E_{N\text{-Ni-atom}}}{N} = \alpha(N) \frac{E_{ds}}{N} - \beta(N) \frac{U_d}{N} + E_0 \quad (\text{R2})$$

For given E_{ds} and U_d , minimizing $E_{\text{per-Ni-atom}}$ with respect to N yields the optimal wavelength N or equivalently optimal wavevector \mathbf{q} . If we know the analytical expression of $\alpha(N)$ and $\beta(N)$, we may find the minimum by taking the derivative $\frac{\partial E_{\text{per-Ni-atom}}}{\partial N} = 0$. However, due to interaction, the analytical expressions of $\alpha(N)$ and $\beta(N)$ are not known. Therefore we numerically calculate $E_{N\text{-Ni-atom}}$ as a function of N for different E_{ds} and U_d . The results are shown in Fig. 5(c) and Fig. 5(d) in the main text.

We note that for a reasonable range of U_d , when $E_{ds} < 0.8$ eV, the optimal wavevector is $\mathbf{q} = (\frac{1}{3}, 0, 0)$; when $E_{ds} \geq 0.8$ eV, the optimal wavevector becomes $\mathbf{q} = (\frac{1}{4}, 0, 0)$. But when $E_{ds} \geq 0.8$ eV, the checkerboard antiferromagnetic (AFM) state has lower energy than the charge ordered state (see Fig. 5(b) in the main text). This means that the CO with a wavevector of $\mathbf{q} = (\frac{1}{4}, 0, 0)$ can not be observed. Instead, it gives way to the AFM state.

Finally, we mention in passing that Eq. (R2) also shows that for a fixed N , increasing U_d (increasing E_{ds}) favors (disfavors) the CO state, which is supported by our numerical

calculations (Fig. 3(b) and Fig. 5(b) in the main text).

In the revised version, we add text on Pages 13 and 14 (Lines 250-264, 277-280) to provide an intuitive picture on this point.

5) Is it possible to quantify the relative importance of the pure electronic vs electron-phonon coupling mechanism for the CDW formation?

We thank the reviewer for this good suggestion. One possible way to quantify the relative importance of electron-electron versus electron-phonon coupling mechanism for the charge ordering is to compare the total energy gain. The charge-transfer mechanism revealed in our work is purely due to repulsive electron-electron interaction. In the effective 2-orbital model, we find that the charge transfer from the correlated Ni- $d_{x^2-y^2}$ orbital to conduction bands yields a total energy gain of about 50 meV/f.u. (see Fig. 3(b) in the main text). If the charge order or charge density wave is driven by electron-phonon coupling, a direct consequence is lattice distortions, which also lower the total energy. Using our effective 2-orbital model, we find that even if we allow lattice distortions in the calculations, the distortions caused by the movement of oxygen atoms can only yield a total energy gain of less than 5 meV/f.u., one order of magnitude smaller (see Fig. 4 in the Supplementary Information). By comparing the total energy gain, we can argue that electron-electron interaction plays a more important role than electron-phonon coupling for the charge order formation in infinite-layer nickelates. In fact, as is demonstrated in our calculations, with a fixed lattice structure, pure electron-electron interaction alone is sufficient to stabilize the charge ordered state in infinite-layer nickelates.

In the revised version, we add and modify text on Page 14 (Lines 286-294 and Lines 296-301) to explain this point.

Response to Reviewer #3

We thank the reviewer for the comments “This is a well written manuscript that merits publication somewhere.” and “The scenario proposed by the authors is intriguing.”.

1) **Problem1** is that there are many other theory papers already published, or in preprint form, circulating as we speak that arrive to the same conclusions using other scenarios. Given the wavevector of the experiments, certainly theorists will find the way to get that result via various channels. Some other authors emphasize phonons for example and also find results matching experiments. How can we tell them apart? Not all of them can appear in *Nat Comm*.

We thank the reviewer for this insightful comment. Indeed there are a few beautiful works emphasizing the importance of phonon and electron-phonon coupling (EPC) for the charge density waves (CDW) in infinite-layer nickelates [15–17]. The mechanism of electron-phonon-driven CDW has been intensively studied in weakly or moderately correlated systems such as transition metal dichalcogenides [18]. A direct consequence of this mechanism is that a characteristic phonon of the ideal crystal structure becomes soft around the CDW transition temperature and the emergence of CDW accompanies a periodic-lattice-distortion, which lowers the total energy [19, 20]. The vibrational mode of characteristic phonon should match the pattern of periodic-lattice-distortion observed at low temperatures [19, 20]. The magnitude of the periodic-lattice-distortion should be such that the resulting energy gain is comparable to the CDW transition temperature [18].

In our scenario, we perform the calculations on the ideal crystal structure of NdNiO_2 and do not need any lattice distortions to stabilize the charge order (CO) in infinite-layer nickelates. Here the driving force for the CO is a charge transfer from a correlated $\text{Ni-}d_{x^2-y^2}$ orbital to conduction bands near the Fermi level. A direct consequence from our scenario is a mixed valence state of Ni that is characterized by a $\text{Ni}^{1+}\text{-Ni}^{2+}\text{-Ni}^{1+}$ stripe pattern. Furthermore, holes are localized in the unidirectional Ni^{2+} chain. Our scenario applies to strongly correlated systems because our calculation clearly shows that the Hubbard U on Ni- d orbitals plays a crucial role.

In addition, we also study possible lattice distortions in the CO state (see Note 5 and Fig. 4 in the Supplementary Information). Our calculations find that even if we allow structural distortions to occur, the distortions in the CO state are minute (0.05 Å movement of oxygen atoms) and the energy gain is less than 5 meV/f.u., substantially smaller than

the energy gain produced by the charge transfer from Ni- d orbital to near-Fermi-level conduction bands (about 50 meV/f.u., see Fig. 3(b) in the main text). Considering that the CO onset is about room temperature in undoped infinite-layer nickelates [21–23], our results imply that the CO in infinite-layer nickelates is more likely of electronic origin, while structural distortions driven by electron-phonon interaction may play a secondary role.

We note that CO in strongly correlated materials is still under intensive investigation [24, 25]. For examples, cuprates also exhibit CO phenomena [11, 12]. While strong EPC may exist in cuprates, there is no clear signature that EPC is essential to the formation of the CO state [26]. Recent experiments have established that infinite-layer nickelates are a new class of strongly correlated electron system [21, 27]. A natural question is whether repulsive electron-electron interaction alone might be sufficient to drive the CO state in infinite-layer nickelates. That is the primary motivation of our work. Here we not only demonstrate a charge-transfer mechanism (not involving electron-phonon interaction and lattice distortions) to induce CO in infinite-layer nickelates, but more importantly the charge transfer is associated with conduction bands near the Fermi level. This is precisely due to the multi-orbital nature of nickelate electronic structure. Our charge-transfer mechanism is absent in cuprates because conduction bands of cuprates (such as those derived from La- d /Sr- d orbitals) are far from the Fermi level.

In summary, if the CO in infinite-layer nickelates arises from our charge transfer mechanism, it will exhibit a mixed valence state of Ni, a characteristic Ni¹⁺-Ni²⁺-Ni¹⁺ stripe pattern, as well as a unidirectional Ni²⁺ chain in which holes are localized. If the CO or CDW is driven by phonon and electron-phonon coupling, it will exhibit a periodic-lattice-distortion whose pattern should match the vibrational mode of a characteristic phonon that gets soft around the CDW transition temperature. The energy gain from the periodic-lattice-distortion should be comparable to the CDW transition temperature. In our reply to comment #3 (see below), we will provide more discussion on various experimental techniques that may find those fingerprints and distinguish CO of different origins.

In the revised version, we add text on Pages 15 and 16 (Lines 329-361) to discuss how

to distinguish different mechanisms for the CO state. We also add and modify text on Page 14 (Lines 286-301) to discuss possible lattice distortions in the CO state and their physical effects.

2) Problem2 is on page 7, sentence “Next, we consider three competing states ...”. This means that the authors focused on only three states. Considering that the PM state is unlikely to be stable in this region of parameters, essentially it is just a competition between two states. So, we do not know whether the true ground state of the effective model is the CO state that the authors are focusing on. It is just the best among three. Basically it is a variational calculation.

We thank the reviewer for this important comment.

First, in Fig. 3 in the main text, we compare the total energy of a uniform paramagnetic state (PM), a checkerboard antiferromagnetic state (AFM) and the $\mathbf{q} = (\frac{1}{3}, 0, 0)$ charge ordered state (CO). The reason that we first focus on these three states is to elucidate the charge-transfer mechanism for the CO state. As Fig. 4 in the main text shows, using the AFM state as the reference state, the charge transfer from the Ni- $d_{x^2-y^2}$ orbital to conduction bands causes a kinetic energy cost because the transferred electron needs to overcome the charge transfer energy E_{ds} ; but in the meantime, the charge transfer also yields a potential energy gain because the charges on one Ni site are depleted, which reduces the local Coulomb repulsion (proportional to U_d). When the potential energy gain outweighs the kinetic energy cost, the CO state becomes more stable than the AFM state. This naturally explains why the CO state rather than the AFM state is observed in infinite-layer nickelates despite a substantial super-exchange found in experiment [27, 28].

Next we study CO states with different wavevectors. In Fig. 5(c) and 5(d) in the main text, we compare a number of CO states with a wavevector $\mathbf{q} = (\frac{1}{N}, 0, 0)$ where $N = 2, 3, 4, 5$. We find that using the DFT fitted value of $E_{ds} = 0.7$ eV, for a reasonable range of U_d , the total energy minimum occurs at a wavevector of $\mathbf{q} = (\frac{1}{3}, 0, 0)$. Furthermore, the variation of CO total energy with its wavelength (i.e. simulation cell size N) is smooth and monotonic when $N \geq 3$ (see Fig. 5(c) in the main text). Therefore we do not expect an irregular fluctuation in the CO total energy when N is larger than 5.

FIG. R4: (a) Schematic of various competing states. The black box is the simulation cell. We study five different charge ordered (CO) states (labelled as CO-I to CO-V), a checkerboard antiferromagnetic state (AFM) and a uniform paramagnetic state (PM). In the CO state, the purple ball highlights the Ni^{2+} site where the charge transfer occurs and a hole is localized. (b) Comparison of total energy of various competing states calculated by the low-energy effective model. For a reasonable range of U_d , the most energetically favorable state is $\mathbf{q} = (\frac{1}{3}, 0, 0)$ charge ordered state (i.e. CO-II), which is highlighted in dark blue.

We agree with the reviewer that if we know how the total energy of CO state depends on wavevector $E_{\text{CO}} = E_{\text{CO}}(\mathbf{q})$, then solving \mathbf{q} from the equation $\nabla_{\mathbf{q}} E_{\text{CO}}(\mathbf{q}) = 0$ yields the optimal wavevector and the local minimum, which is a variational calculation. However, due to interaction, the analytical expression of $E_{\text{CO}}(\mathbf{q})$ is not known. Furthermore, in an interacting lattice model, many different orders can be stabilized in the calculations (such as CO and AFM) and they are not necessarily connected to each other in a continuous way. Therefore in order to find the ground state, we have to compare as many known states as possible (within the computational capability).

In Fig. R4(a), we summarize all the competing states in this study. In addition to PM, AFM and $\mathbf{q} = (\frac{1}{N}, 0, 0)$ CO ($N = 2, 3, 4, 5$), in the revised version we perform additional

calculations and test a different CO state which can be accommodated by the available simulation cell (labelled as CO-V in Fig. R4(a)). Fig. R4(b) compares the total energy of these competing states. We find that for a reasonable range of U_d , $\mathbf{q} = (\frac{1}{3}, 0, 0)$ CO (labelled as CO-II) is the most energetically favorable one among all the states considered.

In the revised version, we add text on Page 15 (Lines 311-320) to explain this important point. We also add a new section (Note 8) in the Supplementary Information about the energetics of all the competing states.

3) Problem3 is that I do not see clear fingerprint predictions of this state so that experiments can be used to distinguish among the many possibilities in the literature that have appeared in print or are about to appear in print.

We thank the reviewer for this important comment. As we mentioned previously, one clear fingerprint prediction of our scenario is the mixed valence state of Ni atoms in the charge ordered state and a characteristic $\text{Ni}^{1+}\text{-Ni}^{2+}\text{-Ni}^{1+}$ stripe pattern. In experiment, one may probe this mixed valence state of Ni either by x-ray absorption spectroscopy (XAS) or cross-sectional electron energy loss spectroscopy (EELS) [9]. XAS can measure the average Ni valence, whose spectrum in the charge ordered state should be distinct from that of pure Ni^{1+} or Ni^{2+} . A more informative measurement is the cross-sectional EELS, which may provide a spatially resolved spectrum of Ni^{1+} and Ni^{2+} columns in the charge ordered state. In addition, scanning tunneling microscopy (STM) may image the localized holes in the unidirectional Ni^{2+} chain [9].

On the other hand, if the charge order or charge density wave (CDW) in infinite-layer nickelates is driven by strong electron-phonon coupling, a soft phonon should appear around the CDW transition temperature and a perodic-lattice-distortion should emerge below the CDW temperature. In experiment, inelastic x-ray scattering or neutron scattering can measure temperature dependence of phonon dispersion and identify soft phonons, if they exist around CDW temperature [20]. The low-temperature perodic-lattice-distortion can be probed either by x-ray diffraction (XRD) or by transmission electron microscopy (TEM). Emergence of perodic-lattice-distortion lowers the crystal symmetry and thus changes the diffraction pattern in XRD measurements [29]. TEM, in particular with the recent developments of electron ptychography [30], can achieve atomic-resolution imaging,

enabling visualization of periodic-lattice-distortion in real space and accurate measurements of structural distortions. Using the lattice distortions found from TEM measurements, one can estimate the energy gain from the periodic-lattice-distortion by performing first-principles calculations and then compare the energy gain to the experimental CDW transition temperature.

The experimental observation of “a mixed valence state of Ni/localized holes on unidirectional Ni^{2+} chain” versus “soft phonon/periodic-lattice-distortion” may distinguish different origins of charge order in infinite-layer nickelates. We hope that the two sets of experiments outlined above, as well as other experimental techniques, will identify the true underlying mechanism among various theoretical proposals.

In the revised version, we add text on Pages 15 and 16 (Lines 329-361) to elaborate this important point. In addition, we also change the title from “The origin of the charge order in infinite-layer nickelates” to “An electronic origin of charge order in infinite-layer nickelates”.

-
- [1] X. Wan, V. Ivanov, G. Resta, I. Leonov, and S. Y. Savrasov. Exchange interactions and sensitivity of the Ni two-hole spin state to Hund’s coupling in doped NdNiO_2 . *Phys. Rev. B* **103**, 075123 (2021).
 - [2] Y. Wang, C.-J. Kang, H. Miao, and G. Kotliar. Hund’s metal physics: From SrNiO_2 to LaNiO_2 . *Phys. Rev. B* **102**, 161118 (2020).
 - [3] C.-J. Kang and G. Kotliar. Optical Properties of the Infinite-Layer $\text{La}_{1-x}\text{Sr}_x\text{NiO}_2$ and Hidden Hunds Physics. *Phys. Rev. Lett.* **126**, 127401 (2021).
 - [4] A. Kreisel, B. M. Andersen, A. T. Rømer, I. M. Eremin, and F. Lechermann. Superconducting Instabilities in Strongly Correlated Infinite-Layer Nickelates. *Phys. Rev. Lett.* **129**, 077002 (2022).
 - [5] H. Chen, C. Xia and H. Liu, Model comparison for infinite-layer nickelates, in preparation (2023).
 - [6] H. Chen, A. Hampel, J. Karp, F. Lechermann, and A. J. Millis. Dynamical Mean Field Studies of Infinite Layer Nickelates: Physics Results and Methodological Implications. *Frontiers*

- in *Physics* **10**, 835942 (2022)
- [7] Y. Shen, J. Sears, G. Fabbris, J. Li, J. Pellicciari, I. Jarrige, X. He, I. Božović, M. Mitrano, J. Zhang, et al.. Role of Oxygen States in the Low Valence Nickelate $\text{La}_4\text{Ni}_3\text{O}_8$. *Phys. Rev. X* **12**, 011055 (2022).
- [8] Y.-F. Jiang, J. Zaanen, T. P. Devereaux, and H.-C. Jiang. Ground state phase diagram of the doped Hubbard model on the four-leg cylinder. *Phys. Rev. Research* **2**, 033073 (2020).
- [9] R. Comin and A. Damascelli. Resonant X-Ray Scattering Studies of Charge Order in Cuprates. *Annual Review of Condensed Matter Physics* **7**, 369-405 (2016).
- [10] G.-M. Zhang, Y.-F. Yang, and F.-C. Zhang. Self-doped Mott insulator for parent compounds of nickelate superconductors. *Phys. Rev. B* **101**, 020501 (2020).
- [11] J. Tranquada, B. Sternlieb, J. Axe, Y. Nakamura, and S. Uchida, Evidence for stripe correlations of spins and holes in copper oxide superconductors. *Nature* **375**, 561 (1995).
- [12] J. Hoffman, E. Hudson, K. Lang, V. Madhavan, H. Eisaki, S. Uchida, and J. Davis. A Four Unit Cell Periodic Pattern of Quasi-Particle States Surrounding Vortex Cores in $\text{Bi}_2\text{Sr}_2\text{CaCu}_2\text{O}_{8+\delta}$. *Science* **295**, 466 (2002).
- [13] M. Hepting, D. Li, C. J. Jia, H. Lu, E. Paris, Y. Tseng, X. Feng, M. Osada, E. Been, Y. Hikita, et al. Electronic structure of the parent compound of superconducting infinite-layer nickelates. *Nature Materials* **19**, 381-385 (2020).
- [14] B.-X. Zheng, C.-M. Chung, P. Corboz, G. Ehlers, M.-P. Qin, R. M. Noack, H. Shi, S. R. White, S. Zhang, and G. K.-L. Chan. Stripe order in the underdoped region of the two-dimensional Hubbard model. *Science* **358**, 1155-1160 (2017).
- [15] X. Sui, J. Wang, X. Ding, K.-J. Zhou, L. Qiao, H. Lin, and B. Huang. Charge order from the local Coulomb repulsion in undoped infinite-layer nickelates. Preprint at <https://arxiv.org/abs/2204.12208>. (2022).
- [16] R. Zhang, C. Lane, J. Nokelainen, B. Singh, B. Barbiellini, R. S. Markiewicz, A. Bansil, and J. Sun. Fingerprints of nematicity and competing orders in the infinite-layer nickelate. Preprint at <https://arxiv.org/abs/2207.00184> (2022).
- [17] A. A. C. Alvarez, L. Iglesias, S. Petit, W. Prellier, M. Bibes, and J. Varignon. Charge ordering as the driving mechanism for superconductivity in rare-earth nickel oxides. Preprint at <https://arxiv.org/abs/2211.04870> (2022).
- [18] S. Manzeli, D. Ovchinnikov, D. Pasquier, O. V. Yazyev, and A. Kis. 2D transition metal

- dichalcogenides. *Nature Reviews Materials* **2**, 17033 (2017).
- [19] M. Holt, P. Zschack, H. Hong, M. Y. Chou, and T.-C. Chiang. X-Ray Studies of Phonon Softening in TiSe_2 . *Phys. Rev. Lett.* **86**, 3799 (2001).
- [20] F. Weber, S. Rosenkranz, J.-P. Castellan, R. Osborn, R. Hott, R. Heid, K.-P. Bohnen, T. Egami, A. H. Said, and D. Reznik. Extended Phonon Collapse and the Origin of the Charge-Density Wave in $2H\text{-NbSe}_2$. *Phys. Rev. Lett.* **107**, 107403 (2011).
- [21] M. Rossi, M. Osada, J. Choi, S. Agrestini, D. Jost, Y. Lee, H. Lu, B. Y. Wang, K. Lee, A. Nag, et al.. A broken translational symmetry state in an infinite-layer nickelate. *Nat. Phys.* **18**, 869 (2022).
- [22] G. Krieger, L. Martinelli, S. Zeng, L. E. Chow, K. Kummer, R. Arpaia, M. Moretti Sala, N. B. Brookes, A. Ariando, N. Viart, et al. Charge and Spin Order Dichotomy in NdNiO_2 Driven by the Capping Layer. *Phys. Rev. Lett.* **129**, 027002 (2022).
- [23] C. C. Tam, J. Choi, X. Ding, S. Agrestini, A. Nag, B. Huang, H. Luo, M. García-Fernández, L. Qiao, and K.-J. Zhou. Charge density waves in infinite-layer NdNiO_2 nickelates. *Nat. Mater.* **21**, 1116-1120 (2022).
- [24] J. M. Tranquada. Cuprate superconductors as viewed through a striped lens. *Advances in Physics* **69**, 437-509 (2020).
- [25] D. F. Agterberg, J. S. Davis, S. D. Edkins, E. Fradkin, D. J. Van Harlingen, S. A. Kivelson, P. A. Lee, L. Radzihovsky, J. M. Tranquada, and Y. Wang. The Physics of Pair-Density Waves: Cuprate Superconductors and Beyond. *Annual Review of Condensed Matter Physics* **11**, 231 (2020).
- [26] X. Zhu, Y. Cao, J. Zhang, and J. Guo. Classification of charge density waves based on their nature. *PNAS* **112(8)**, 2367 (2015).
- [27] H. Lu, M. Rossi, A. Nag, M. Osada, D. F. Li, K. Lee, B. Y. Wang, M. Garcia-Fernandez, S. Agrestini, Z. X. Shen, et al. Magnetic excitations in infinite-layer nickelates. *Science* **373**, 213-216 (2021).
- [28] D. Zhao, Y. B. Zhou, Y. Fu, L. Wang, X. F. Zhou, H. Cheng, J. Li, D. W. Song, S. J. Li, B. L. Kang, et al. Intrinsic Spin Susceptibility and Pseudogaplike Behavior in Infinite-Layer LaNiO_2 . *Phys. Rev. Lett.* **126**, 197001 (2021).
- [29] C. Holder and R. Schaak. Tutorial on Powder X-ray Diffraction for Characterizing Nanoscale Materials. *ACS Nano* **13**, 7359 (2019).

- [30] Z. Chen, S. Jiang, Yi, Yu-Tsun, M. E. Holtz, M. Odstreil, M. Guizar-Sicairos, I. Hanke, S. Ganschow, D. G. Schlom, and D. A. Muller. Electron ptychography achieves atomic-resolution limits set by lattice vibrations. *Science* **372**, 826 (2021).

REVIEWERS' COMMENTS

Reviewer #1 (Remarks to the Author):

This manuscript addressed a charge ordering in the NdNiO₂ that is a superconductor when hole-doping, as happened in the high T_c cuprate CaCuO₂.

The authors suggest that using the DMFT scheme inclusion of on-site Coulomb repulsion leads to the charge order of Ni(1+)-Ni(2+)-Ni(1+) with $q=(1/3,0,0)$ by a charge transfer among the ions in a relatively large U region.

This result was also reproduced and analyzed by using a two-band effective model.

Although no experimental evidence has been reported yet, this result would be of interest and could be helpful to understand the mechanism of the superconductivity.

Additionally, this manuscript is well revised.

Thus, this manuscript can be recommended to be published.

The referee has one more question.

This charge transfer observed here seems to be very similar to that of the DFT+U scheme in the high U region as shown in Ref. [2]. Thus, the referee wonders if the singlet state in Ref. [2] appears in the Ni(2+) ions. The orbital resolved spectra would provide more useful information.

Reviewer #2 (Remarks to the Author):

The authors have made an effort of clarifying some important points of their work, also in relation to previous works. The new version of the manuscript has improved with these clarifications, even if it remains quite technical. I think that the paper deserves publication somewhere.

Reviewer #3 (Remarks to the Author):

The authors have partially answered my concerns, particularly my critique that they had compared only 3 states: PM, CO $1/3$ and AFM. They explained that they also used CO $1/2$, $1/4$, $1/5$, and moreover in their response to my report, they have another charge ordered state CO-V. Thus, leaving aside PM, they have basically compared 6 states: AFM, and five CO states, finding overall that the CO $1/3$ has the lowest energy.

The authors also nicely explained that a CO due to phonons lowers the energy by a much smaller amount than the CO due to electrons that they propose.

In view of these improvements, I am willing to give to the authors the benefit of the doubt, and although with some hesitation I recommend publication if the other referees are also in agreement.

My hesitation arises from the sharpness of the profile $Ni^{1+}-Ni^{2+}-Ni^{1+}$ (I would have expected a more smooth charge profile) as well as the absence of magnetic order in between the stripes. For instance, in the work of Tranquada et al. in the cuprates, they have stripes at $x=1/8$ and in between the lines of holes, they have spin two leg ladders with magnetic moments and a spin gap. Thus, at least in cuprates it is a mixture of CO and magnetic moments forming a nice spin liquid. Thus, I doubt that the results in this paper are the final answer. But it is a starting point.

In summary, from my perspective it is ok to publish but I recommend the editors to focus their attention on the reports of the other referees.

Reply to Reviewers (NCOMMS-22-53047A)

Response to Reviewer #1

We thank the reviewer for the comment that “this manuscript is well revised” and recommending publication of our work.

1) This charge transfer observed here seems to be very similar to that of the DFT+U scheme in the high U region as shown in Ref. [2]. Thus, the referee wonders if the singlet state in Ref. [2] appears in the Ni(2+) ions. The orbital resolved spectra would provide more useful information.

We thank the reviewer for this comment.

In PRB 70, 165109 (2004) (i.e. Ref. [2] in the manuscript), the authors performed DFT+U calculations and studied an antiferromagnetic state of LaNiO₂. They found that when U on Ni- d orbital is sufficiently large (about 8 eV), there is a charge transfer from Ni- $3d$ states to La- $5d$ states, which moves the valence state of Ni from Ni⁺ to Ni²⁺. They also found that in this approximate Ni²⁺ state, one hole occupies the down spin of $d_{x^2-y^2}$ orbital and the other hole occupies the up spin of $d_{3z^2-r^2}$ orbital. Therefore the net moment on each Ni atom becomes very small (about $0.2\mu_B$ per Ni). The authors of PRB 70, 165109 (2004) referred to this state as “an onsite singlet”.

First we need to explain that the authors of PRB 70, 165109 (2004) used a static mean-field theory method (DFT+U) and calculated an antiferromagnetic state. In our work, we use a dynamical mean field theory method (DFT+DMFT). The charge ordered state in our calculations is spin disordered (i.e. not magnetically ordered) but charge and spin fluctuations are fully taken into account. Therefore strictly speaking, we can not directly compare the Ni²⁺ in our charge ordered state to the Ni²⁺ state that is found in PRB 70, 165109 (2004). Yet we do find some similarity between the two cases. Using our 17-orbital model calculations (with $U_{\text{Ni}} = 10$ eV and $J_{\text{Ni}} = 1$ eV), we calculate the orbital resolved spectral functions of Ni²⁺ and show the results in Fig. R1. We find that both

FIG. R1: Orbital resolved spectral function of Ni^{2+} ion in the charge ordered state of NdNiO_2 , calculated using the 17-orbital model with $U_{\text{Ni}} = 10$ eV and $J_{\text{Ni}} = 1$ eV. The blue, red and green curves are the orbital resolved spectral functions projected onto $\text{Ni-}d_{x^2-y^2}$, $\text{Ni-}d_{3z^2-r^2}$ and $\text{Ni-}t_{2g}$ orbitals. The black dashed line is the Fermi level.

$\text{Ni-}d_{x^2-y^2}$ and $\text{Ni-}d_{3z^2-r^2}$ orbitals are close to half-filled, while $\text{Ni-}t_{2g}$ orbitals are almost fully occupied. By integrating the orbital resolved spectral function, we find that for Ni^{2+} ion in the charge ordered state of NdNiO_2 , the occupancies of $d_{x^2-y^2}$ and $d_{3z^2-r^2}$ orbitals are 1.06 and 1.11, respectively. This indicates that each Ni^{2+} e_g orbital has about one hole in it, which is similar to what is found in PRB 70, 165109 (2004).

Response to Reviewer #2

We thank the reviewer for the comment “The new version of the manuscript has improved with these clarifications” and recommending publication of our work.

Response to Reviewer #3

We thank the reviewer for recommending publication of our work.

1) My hesitation arises from the sharpness of the profile $\text{Ni}^{1+}\text{-Ni}^{2+}\text{-Ni}^{1+}$ (I would have expected a more smooth charge profile) as well as the absence of magnetic order in between the stripes. For instance, in the work of Tranquada et al. in the cuprates, they have stripes at $x=1/8$ and in between the lines of holes, they have spin two leg ladders with magnetic moments and a spin gap. Thus, at least in cuprates it is a mixture of CO and magnetic moments forming a nice spin liquid. Thus, I doubt that the results in this paper are the final answer. But it is a starting point.

We thank the reviewer for raising this important comment.

We agree with the reviewer that the stripe states in cuprates are more complicated with both charge order and spin order, as well as real-space segregation of holes and spins [1, 2]. Many-body calculations on single-orbital Hubbard model find such a stripe state with a profile of gradually changing charges and magnetic moments [3].

In our calculations, we explicitly include conduction bands in our modelling (both in the 17-orbital model and in the 2-orbital model). A charge transfer from a portion of Ni atoms to the conduction bands leads to a $\mathbf{q} = (\frac{1}{3}, 0, 0)$ charge ordered state, which is recently observed in infinite-layer nickelates [4–6]. After the charge transfer, the holes on Ni^{2+} atoms become very localized due to strong correlation effects.

We fathom that the sharpness of the $\text{Ni}^{1+}\text{-Ni}^{2+}\text{-Ni}^{1+}$ charge profile found in our calculations is probably because of this particular charge transfer mechanism as well as the short periodicity of the charge order. By contrast, in calculations of the single-orbital Hubbard model or t - J model, the charge modulation arises from the charge transfer between *neighboring correlated sites* with no conduction band involved, which results in a charge order of a longer periodicity and a more gradual change in the charge profile [3, 7, 8]. This issue warrants further studies.

Finally, we completely agree with the reviewer that the study on charge ordering phenomena in infinite-layer nickelates has just begun. We hope that our results may stimulate

further theoretical and experimental work on nickelate superconductors.

- [1] J. Tranquada, B. Sternlieb, J. Axe, Y. Nakamura, and S. Uchida, *Nature* **375**, 561 (1995).
- [2] J. M. Tranquada, *Advances in Physics* **69**, 437 (2020).
- [3] B.-X. Zheng, C.-M. Chung, P. Corboz, G. Ehlers, M.-P. Qin, R. M. Noack, H. Shi, S. R. White, S. Zhang, and G. K.-L. Chan, *Science* **358**, 1155 (2017).
- [4] M. Rossi, M. Osada, J. Choi, S. Agrestini, D. Jost, Y. Lee, H. Lu, B. Y. Wang, K. Lee, A. Nag, et al., *Nat. Phys.* **18**, 869 (2022).
- [5] G. Krieger, L. Martinelli, S. Zeng, L. E. Chow, K. Kummer, R. Arpaia, M. Moretti Sala, N. B. Brookes, A. Ariando, N. Viart, et al., *Phys. Rev. Lett.* **129**, 027002 (2022).
- [6] C. C. Tam, J. Choi, X. Ding, S. Agrestini, A. Nag, B. Huang, H. Luo, M. García-Fernández, L. Qiao, and K.-J. Zhou, *Nat. Mater.* **21**, 1116 (2022).
- [7] Y.-F. Jiang, J. Zaanen, T. P. Devereaux, and H.-C. Jiang, *Phys. Rev. Research* **2**, 033073 (2020).
- [8] H.-C. Jiang, Z.-Y. Weng, and S. A. Kivelson, *Phys. Rev. B* **98**, 140505 (2018).